# Genes associated with cognitive ability and HAR show overlapping expression patterns in human cortical neuron types

Stan L. W. Driessens[1,4], Anna A. Galakhova [1,4], Djai B. Heyer[1,4], Isabel J. Pieterse[1], René Wilbers [1], Eline J. Mertens[1], Femke Waleboer[1], Tim S. Heistek[1], Loet Coenen [1], Julia R. Meijer [1], Sander Idema[2], Philip C. de Witt Hamer [2], David P. Noske[2], Christiaan P. J. de Kock [1], Brian R. Lee[3], Kimberly Smith [3], Jonathan T. Ting[3], Ed S. Lein [3], Huibert D. Mansvelder [1] & Natalia A. Goriounova [1]✉

GWAS have identified numerous genes associated with human cognition but their cell type expression profiles in the human brain are unknown. These genes overlap with human accelerated regions (HARs) implicated in human brain evolution and might act on the same biological processes. Here, we investigated whether these gene sets are expressed in adult human cortical neurons, and how their expression relates to neuronal function and structure. We find that these gene sets are preferentially expressed in L3 pyramidal neurons in middle temporal gyrus (MTG). Furthermore, neurons with higher expression had larger total dendritic length (TDL) and faster action potential (AP) kinetics, properties previously linked to intelligence. We identify a subset of genes associated with TDL or AP kinetics with predominantly synaptic functions and high abundance of HARs.

Our mind functions through the activity of almost a hundred billion neurons and their connections[1]. They form principal building blocks for integrating, processing, and storage of information in the brain and ultimately support cognitive ability. Given the astronomical number of neurons and neuronal connections in the human brain[2], even the slightest changes in the efficiency of single neuron function can translate into large differences in cognition between individuals.

Neuronal networks underlying cognition are under strong genetic control and share common genetic origin with human intelligence[3,4]. Multiple genetic variants associated with intelligence and educational attainment (EA) have been recently identified by genome-wide association studies (GWAS) and mapped to hundreds of candidate genes[5,6]. These genes are implicated in various neuronal functions including synaptic function and plasticity, cell interactions and energy metabolism[5,7] and relate to brain volume[3]. Especially in higher-order

cortical areas, such as middle temporal gyrus (MTG), volume, cortical thickness and cortical activation were shown to be statistically associated with differences in intelligence quotient (IQ) scores in the human population[8–11]. At the same time, similar cortical expansion of higher-order cognitive areas has been linked to human brain evolution and genes associated with accelerated evolution in humans (human accelerated regions, HARs)[12]. The challenge is to understand whether and which of these genes are expressed in human cortical areas and neurons, and how they influence neuronal function and structure not only during development but most importantly in adult mature neurons and networks that engage in cognitive performance.

Genes exert their function at the protein level and are expressed in neurons, thereby shaping neuronal function. Thus, individual genetic variation will express itself first and foremost at the cellular level, and cellular consequences will combine to act at the neuronal

[1]Department of Integrative Neurophysiology, Amsterdam Neuroscience, Center for Neurogenomics and Cognitive Research (CNCR), Vrije Universiteit Amsterdam, De Boelelaan 1085, Amsterdam 1081 HV, the Netherlands. [2]Department of Neurosurgery, Amsterdam UMC Location Vrije Universiteit Amsterdam, De Boelelaan 1117, 1081HV Amsterdam, the Netherlands. [3]Allen Institute for Brain Science, 615 Westlake Ave N, Seattle, WA 98109, USA. [4]These authors contributed equally: Stan L. W. Driessens, Anna A. Galakhova, Djai B. Heyer. ✉e-mail: n.a.goriounova@vu.nl

circuit, influencing brain areas and brain function. Indeed, several morpho-electrical properties of human pyramidal neurons in cortical layers 2 and 3 (L2/L3) from higher-order association areas (such as in MTG) were shown to correlate with measures of intelligence (IQ scores) in human subjects[13,14]. Pyramidal neurons from subjects with higher IQ had larger and more complex dendrites and were better able to maintain fast action potential (AP) kinetics. Longer dendrites endow neurons with larger dendritic surface that can physically harbor more synaptic inputs[2], while fast and stable APs might improve overall processing speed and temporal resolution to process multiple simultaneous inputs[13,15,16]. These morpho-electric properties coincide with those found to be enhanced in human neurons compared to other species: human neurons have 3-fold larger dendritic trees[17] and faster APs than mouse neurons[16]. Are these L2/L3 pyramidal neurons and their structural and functional properties under control of genes associated with human cognition and HARs?

Recent advances in single cell RNA-sequencing make it now possible to combine full genetic fingerprints of single human neurons with the study of their morphological and functional properties by combining patch-clamp recordings with RNA-sequencing (Patch-seq technique)[18,19]. Thus, information on the levels of all expressed genes (RNA transcripts) can be derived from single cells and based on the cell's transcriptomic signature each cell can be assigned to a transcriptomically-defined neuronal type (t-type). Recently, pyramidal neurons in L2/L3 of MTG have been transcriptomically mapped to 5 distinct t-types[18,19]. Some of these types – L3 types CARM1P1 and FREM3 – stand out because of their large and elaborate dendritic trees. Moreover, these neurons can be labeled by neurofilament marker SMI-32 that indicates long-range cortico-cortical projection neurons that are selectively lost in Alzheimer's disease[19,20]. Furthermore, CARM1P1 does not have a homologue in mouse cortex and might be a human-specialized type[19]. The fact that these cells are vulnerable in a disease marked by cognitive decline suggests their importance for human cognition.

Here we leveraged RNA-seq data and Patch-seq data from human neurons to investigate gene expression profiles in human cortical areas and neurons for 3 gene sets: genes associated with IQ score variation (further referred to as IQ genes)[5], genes associated with educational attainment in 1.1 million participants (EA genes)[6] and HAR genes conserved across vertebrates but highly divergent in humans[21]. We hypothesized that these gene sets would be enriched in the cortical areas linked to human intelligence[9,13,14] and language[22] and expanded during human evolution[12] such as the multimodal association area MTG. Next, we focused on human-specialized L3 neuronal types—CARM1P1 and FREM3[18,19]. Furthermore, we asked whether expression levels of these genes associated with cellular phenotypes that explain interindividual differences in intelligence—dendritic size and speed of AP signaling. These neuronal properties are also those that distinguish human from mouse neurons and thus might not only be under control of genes associated with cognition (IQ and EA genes), but also those implicated in human evolution – HAR genes. Finally, we aimed to identify genes most robustly related to these morpho-electrical properties and investigate their biological function.

## Results
### Enriched expression in glutamatergic neurons of human MTG
Several cortical association regions in frontal and temporal lobes of the human brain play a central role in human cognition[9] and show larger evolutionary expansion compared to the cortex of chimpanzees[12]. One such area is the MTG that is a part of the semantic system[22,23] and verbal cognition[14]. We first asked whether IQ, EA and HAR genes are preferentially expressed in this area compared to motor and primary sensory cortical areas that are not part of the higher-order cognitive network (Fig. 1a). We utilized the Allen Cell Types Database[18,24] consisting of single-nucleus RNA sequencing of neuronal

and non-neuronal cells in 6 human cortical areas: MTG, cingulate gyrus (CgC), primary motor cortex (M1), and primary visual (V1), somato-sensory (S1) and auditory (A1) cortices (Fig. 1).

To avoid the possibility that some cell types, classes, or areas have generally higher expression of all genes, gene expression levels of each gene were normalized to the total number of reads in each sample (cell nucleus). We used the count per million (CPM) values that were obtained by dividing the number of reads mapped to each gene of interest by the total number of mapped reads in each cell and multiplying by a million. To investigate the gene expression profiles across areas and neurons, we collapsed the gene expression levels of the associated genes in each gene set into one value: the mean expression level of all associated genes in each individual cell. Furthermore, we excluded a minority of genes from the analysis that had zero expression in more than 95% of cells (based on the smaller Patch-seq dataset of 276 L2/L3 glutamatergic types). Using this criterion, we found that most of these genes were expressed in the glutamatergic neurons: 936 of 1016 IQ genes (92%), 1642 of 1838 EA genes (89%) and 1476 of 1711 HAR genes (86%) (Fig. 1a right).

In the cortex, excitatory glutamatergic pyramidal neurons are the principal neurons that accumulate and integrate synaptic information and pass it on to other cortical and subcortical areas, while inhibitory interneurons locally regulate their activity[2]. We asked whether the expression profiles of the three gene sets were different across cortical areas for the three main classes of brain cells: excitatory glutamatergic neurons, GABAergic inhibitory interneurons and non-neuronal cells.

The high-throughput, single cell RNA-sequencing data were obtained from thousands of individual cells but originated only from a few human donors ($N = 3$ for all cortical areas, $N = 4$ for MTG). To account for nested data, we performed two analyses. In the first analysis, we averaged the mean expression per individual donor and compared gene sets of interest across brain areas, in this analysis donors are considered as statistical units (data points in Fig. 1). In the second analysis, we tested differences in mean expression levels of the gene sets within individual donors, comparing brain areas and cell types for each donor separately, in this case single cell expression levels are considered as statistical units (violins in Fig. S1). These analyses showed similar results: the expression levels of genes from IQ, EA, and HAR gene sets were lower in primary sensory areas (S1, V1, A1), and higher in MTG, CgG and M1, especially in glutamatergic neuron types (Fig. 1b and Fig. S1). In GABAergic and non-neuronal cells, the expression pattern of the three gene sets was more homogenous across brain areas (Fig. 1c, d). Furthermore, when focusing on the MTG, mean expression levels of IQ, EA, and HAR genes were higher in glutamatergic cells than in non-neuronal cells (Fig. 1e, data per cell class is shown for MTG), suggesting a potential larger role of genes associated with cognition in glutamatergic neurons.

### Enriched expression in L3 glutamatergic neuron types
Glutamatergic neuron types in L2/3 of MTG were previously shown to associate with human cognitive function[13,14]. Especially the large L3 neurons in association cortices such as MTG likely support human cognition, as they are selectively lost in Alzheimer's disease[19,20,25]. We asked whether genes associated with human cognition are enriched in these L3 neuron t-types.

The L2/L3 glutamatergic neuron types can be divided into 5 t-types based on their transcriptomic signature[19]. Each type has its preferred location within cortical layers and distinct morpho-physiological properties: the LTK type was strictly localized to L2; the most abundant and diverse FREM3 type was found across L2 and L3, the GLP2R was located in superficial L3, and the deep L3 types – large and branchy CARM1P1 type and slim-shaped COL22A1[19] (Fig. 2a). Similar to the analysis in Fig. 1, we used snRNA-seq data to test whether IQ, EA, and HAR genes were differentially expressed in these L2/L3

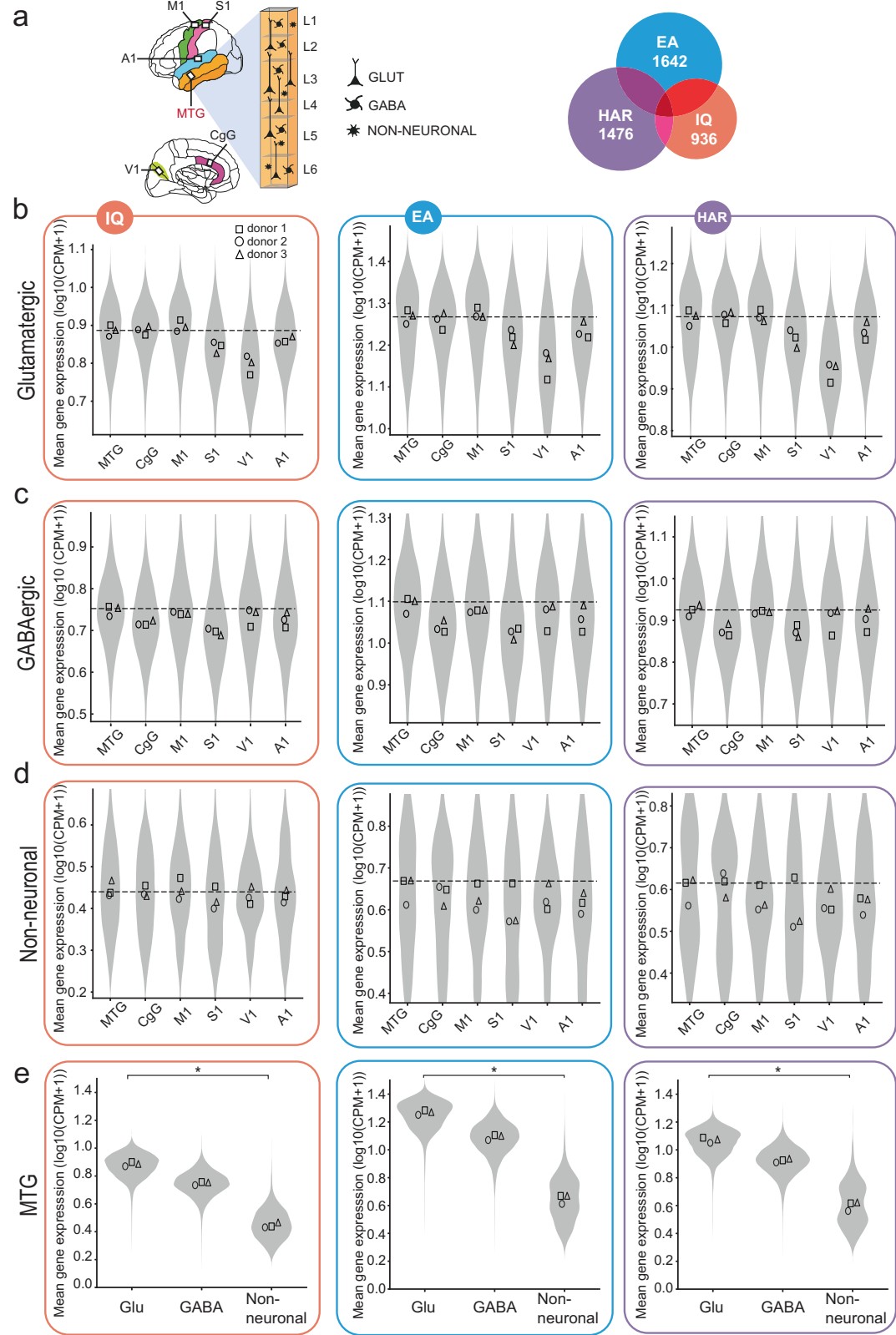

glutamatergic types (Fig. 2b). Because of large variability within the FREM3 type and its location across L2 and L3, we split this type into L2 and L3 subtypes based on their cortical location. We compared mean expression of gene sets across cell types, and analyzed mean expression of gene sets across cell types per donor. Mean gene expression levels of IQ, EA, and HAR gene sets were elevated in L3 FREM3 and CARM1P1 t-types (Fig. 2b).

Following the results from the single cell RNA-seq data, we looked further into L2/L3 glutamatergic cells from MTG using Patch-seq data from our earlier work[19]. Patch-seq combines patch-clamp recording of single neurons with RNA sequencing of nuclear content of the same neuron and thus allows not only access to transcriptomic information but also characterization of cellular physiological and morphological features (Fig. 2c). We first asked whether the results from RNA-seq data

**Fig. 1 | Genes associated with IQ and EA and HAR are enriched in glutamatergic neuron types. a** Schematic representation of the human brain areas from which the Allen Brain snRNA-seq data were analyzed. Venn diagram shows the number of genes expressed in brain cell types for each gene set: genes associated with IQ, educational attainment (EA) and HAR. Within each gene set, single cell RNA-expression data were averaged across genes for each cell. **b** Expression profiles of IQ, EA, and HAR gene sets across brain areas in glutamatergic cells across cortical areas. Here and further: IQ, EA and HAR gene set data are displayed in orange, blue and purple frames, respectively. Violins show the distribution of cellular data of all donors, where each data point is the mean gene expression for each gene set per cell ($N$ = 502-10324 cells, K-W test $p$ values are all significant; see Source Data for N for each cell class and area and K-W test $p$ values for each gene set). Symbols within violins show the median gene expression per human donor (two-sided Friedman test glutamatergic types: $N$ = 3 donors; IQ: F = 13.48, $p$ = 0.0009; EA: F = 13.1, $p$ = 0.0018; HAR: F = 13.1, p = 0.0018). Dashed line marks the median gene expression in MTG. **c** The expression levels in GABAergic and **d** non-neuronal cells across brain areas (two-sided Friedman test GABA: IQ: F = 10.81, $p$ = 0.0247, EA: F = 8.905, $p$ = 0.0905, HAR: F = 7.381, $p$ = 0.2014; Non-neuronal: IQ: F = 4.524, $p$ = 0.534, EA: F = 4.714, $p$ = 0.5108, HAR: F = 6.619, $p$ = 0.2755). **e** Expression of IQ, EA, and HAR genes in cell classes within MTG (two-sided Friedman test: $N$ = 3 donors; IQ: F = 6, $p$ = 0.0278; EA: F = 6, $p$ = 0.0278; HAR: F = 6, $p$ = 0.0278). Asterisks indicate the result of a two-sided Friedman test of donor data here and further: *$p$ < 0.05; **$p$ < 0.01; ***$p$ < 0.001. Complete N and statistical results are provided as a Source Data file.

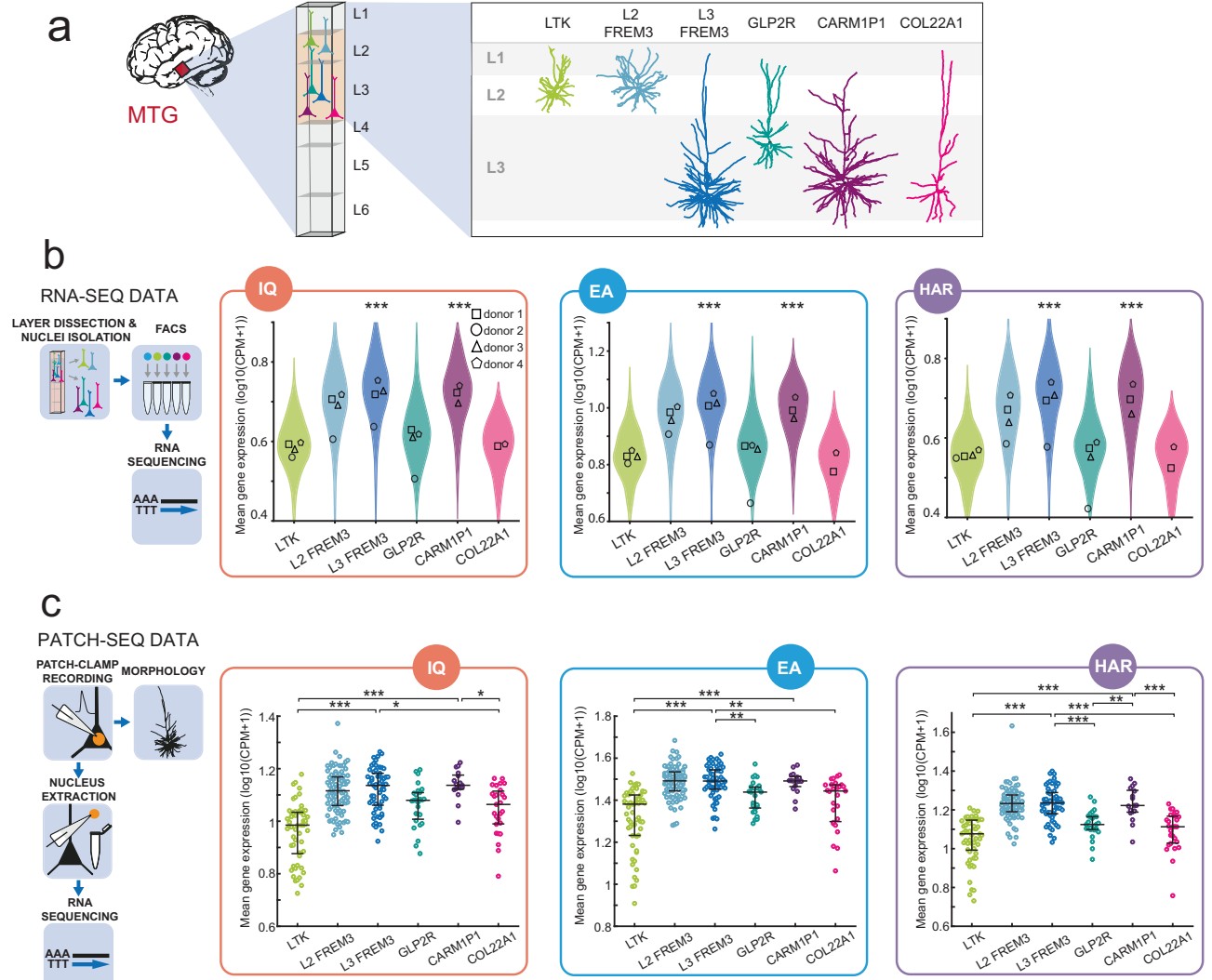

**Fig. 2 | Genes associated with IQ and EA and HAR are enriched in L3 FREM3 and CARM1P1 t-types. a** Examples of reconstructed dendritic morphologies of excitatory L2/L3 human t-types from MTG area. **b** Schematic represents the workflow of RNA-seq data collection. Expression of IQ, EA and HAR gene sets across MTG L2/L3 glutamatergic neuron types. Violins show the distribution of snRNA-seq data for all donors, where each data point is the mean gene expression of each gene set per cell (N (neurons): LTK = 666; L2 FREM3 = 605; L3 FREM3 = 1336; GLP2R = 148; CARM1P1 = 172; COL22A1 = 41; Kruskal–Wallis (KW) test; IQ gene set; $P$ = 1.18 × 10$^{-16}$; EA gene set: $P$ = 4.6 × 10$^{-18}$; HAR gene set: $P$ = 2.06 × 10$^{-26}$. Symbols within violins show the median gene expression per human donor ($N$ = 3; two-sided Friedman test of donor data: IQ gene set: F = 9.714, $p$ = 0.0083; EA gene set: $F$ = 10, $p$ = 0.0014; HAR gene set: F = 9,429, $p$ = 0.0167). Asterisks above t-types indicate that this t-type was significantly different in K-W test post-hoc comparisons to LTK, GLP2R, and COL22A1 t-types and represent the least significant difference. **c** Schematic represents the collection workflow of Patch-seq data. Plots show mean gene expression levels of IQ, EA and HAR gene sets in glutamatergic t-types collected by Patch-seq. For Patch-seq here and further: data points represent data of individual neurons; color code for the t-types is the same across all figures; black horizontal lines are the median values and vertical lines show the interquartile range (IQR). N (neurons) for each group: LTK = 57; L2 FREM3 = 90; L3 FREM3 = 58; GLP2R = 27; CARM1P1 = 17; COL22A1 = 27. K-W test; IQ gene set: $P$ = 1.2 × 10$^{-16}$; EA gene set: $P$ = 4.7 × 10$^{-18}$; HAR gene set: $P$ = 2.1 × 10$^{-26}$. Asterisks indicate the significance of K-W test post-hoc comparisons for only CARM1P1 and L3 FREM3 types. Plots indicate median (middle line) and 25th and 75th percentiles (whiskers). Complete N and statistical results are provided as a Source Data file.

can be confirmed in the much smaller Patch-seq dataset[19,26]. We compared the glutamatergic t-types and found that Patch-seq data showed similar enrichment of mean gene expression levels of IQ, EA and HAR gene sets in L3 t-types: L3 FREM3 and CARM1P1 had higher gene expression levels compared to L2 t-type LTK in all three gene sets (Fig. 2c). The low-throughput Patch-seq data with complete physiological, morphological and transcriptomic profile of each cell was collected from 57 donors. To account for potentially nested data, similar to the RNA-seq data analysis, we analyzed the Patch-seq data averaged per individual donor (Fig. S2). We obtained similar results, confirming that L3 human-specific types, in particular, show the highest expression of genes related to human cognition and brain evolution.

## Convergent expression patterns in neurons with large dendrites

Neurons connect to each other with synaptic contacts on their dendrites; the larger the dendrites the more physical space there is for possible connections. Indeed, larger pyramidal neurons receive more synaptic inputs[27,28]. In addition, larger dendrites of L2/L3 glutamatergic neurons are one of the most distinctive features of human brain evolution[17,28]. At the same time, interindividual variability in total dendritic length in MTG was linked to IQ scores in human subjects[29] and measures of functional integration within the global brain network[29,30]. Such differences might be under control of genes associated with human cognition (IQ and EA) and implicated in human evolution (HAR).

To address this, we computed total dendritic length (TDL) for each neuron from our Patch-seq data and compared it between t-types. Deep L3 neuron types FREM3 and CARM1P1 had the largest TDL of all types and their dendrites were 2 to 3-fold longer compared to more superficially positioned L2/L3 types (Fig. 3a). If genes associated with human cognition influence the dendritic structure of neurons, we hypothesized that larger neurons express more of those genes. To test this, we correlated TDL to the mRNA expression levels of each gene in the three gene sets of interest (IQ, EA and HAR). Next, we ranked the genes according to the significance of the correlation (p value) to TDL, after correcting for multiple correlations with false discovery rate (FDR) of 0.05. The heatmaps in Fig. 3b–d show the significantly correlating genes per neuron and the gradients in their mRNA expression levels for each gene set. The majority of correlating genes showed a positive association of expression with TDL (top heatmaps in Fig. 3b–d), but a few genes showed a negative association of expression with TDL in all three gene sets (bottom heatmaps in Fig. 3b–d). Since TDL changes in a gradient from low values in L2 types to high values in L3 types (Fig. 3b–d, color-coded neuron t-types ranked by TDL are shown above the heatmaps), we expected that the correlating genes would be differentially expressed across t-types. Indeed, average expression of positively correlated genes was the highest in L3 FREM3 and CARM1P1 types in all gene sets, while a smaller number of negatively correlating genes showed the lowest expression in these types (Fig. 3b–d, right plots). In total, we found that TDL correlated significantly to gene expression levels in a subset of 37 genes from IQ, 68 from EA and 85 from HAR gene sets (Fig. 3e). Notably, the TDL-correlated genes had a larger overlap between the gene sets: many correlated genes belonged to two or all three gene sets (Fig. 3e, detailed information on these genes is listed in Table S1). These results identify a subset of genes from IQ, EA and HAR gene sets that are up- and downregulated in large L3 neuron types. Their differential expression in neurons with small and large dendrites may suggest that neuronal dendritic structure is regulated by genes that are related to both human cognition and human brain evolution.

## Convergent expression patterns in neurons with fast AP kinetics

A critical requirement for fast processing in neurons and neuronal networks is rapid action potential (AP) generation that is stable during repetitive firing, i.e., the speed of the rising flank of the AP does not slow down but remains fast when neurons are engaged in cortical processing[13,16]. AP initiation and rise speed helps neurons to react faster to rapidly changing subthreshold membrane potential changes induced by synaptic inputs and convert these inputs into AP output[13,16,30,31]. The faster AP initiation, the larger the information content (bandwidth) that can be encoded in AP output timing, thereby enabling large bandwidth information processing in neuronal networks[30,31]. Human neurons with large dendrites receive many more synaptic inputs than human neurons with small dendrites[2,27], leading to more temporal fluctuations in subthreshold membrane potential changes induced by ongoing synaptic transmission. Fast AP initiation kinetics are thereby particularly important for large pyramidal neurons that receive more synaptic inputs to be able to encode a larger information space into AP output timing. Large neurons with fast AP initiation mechanisms can process and relay much more information[16]. This is relevant for cognition, since neurons are the basic units of cortical information processing. Indeed, in human subjects with higher IQ scores, AP rise speed of pyramidal neurons remains fast during repetitive firing and shows a positive correlation with IQ scores[13,14]. Especially in larger L3 neuronal types, AP speed might be critical to reliably encode multiple simultaneous inputs on their vast dendrites.

We next asked whether AP rise speed is higher in L3 neurons and relates to the expression levels of genes associated with cognition. To test this, we recorded AP firing in neurons using patch-clamp in response to current injections (Fig. 4a). AP rise speed was defined as the maximum speed (derivative) of the rising phase of the AP (Fig. 4b). We observed the fastest AP rise speeds in CARM1P1 type followed by L3 FREM3 neurons (Fig. 4c).

Then we explored which genes from the IQ, EA, and HAR gene sets correlate with AP rise speed in human neuron t-types. To test this, we correlated AP rise speed to the expression levels of each gene in the three gene sets (IQ, EA, and HAR). Similar to the analysis in Fig. 3, we ranked the genes according to the significance of the correlation (p values) to AP rise speed, and corrected for multiple correlations with false discovery rate (FDR) of 0.05. The heatmaps in Fig. 4d–f show the significantly correlating genes per neuron with AP rise speeds and the gradients in their mRNA expression levels for each gene set. We found that the majority of significantly correlating genes showed a positive correlation of expression with AP rise speed (top heatmaps in Fig. 4d–f). A smaller group of genes showed a negative correlation of expression with AP rise speed (bottom heatmaps in Fig. 4d–f). To test whether expression of positively and negatively correlating genes was different in the different neuron t-types (Fig. 4d–f, color-coded neuron t-types ranked by AP rise speed are shown above the heatmaps), we compared the mean expression of these genes in the cell types (Fig. 4d–f, right plots). The average expression of positively correlated genes was the highest in L3 FREM3 and CARM1P1 types in all gene sets, while negatively correlating genes showed the lowest expression in these types (Fig. 4d–f, right plots, see Table S1 for detailed information on TDL- and AP-correlating genes). In total, we found that AP rise speed correlated significantly to gene expression levels in a subset of 26 genes from IQ, 49 from EA and 81 from HAR gene sets (Fig. 4g). As in the subset of TDL-correlated genes, AP-correlated genes also showed a larger overlap between the gene sets: many correlated genes belonged to two or all three gene sets (gene names are listed in Fig. 4g). Thus, faster AP rise speeds in L3 neuron t-types might be under control of a subset of overlapping genes from IQ, EA and HAR gene sets.

## Genes related to neuronal structure and function are shared across gene sets

Our results show that cellular phenotypes associated with intelligence - dendritic size and AP rise speed – may be influenced by genes associated with cognitive ability. Are these genes also related to human evolution? For this, we focused on the subset of IQ, EA, and HAR genes that significantly correlated with either TDL (42 genes listed in Fig. 3e)

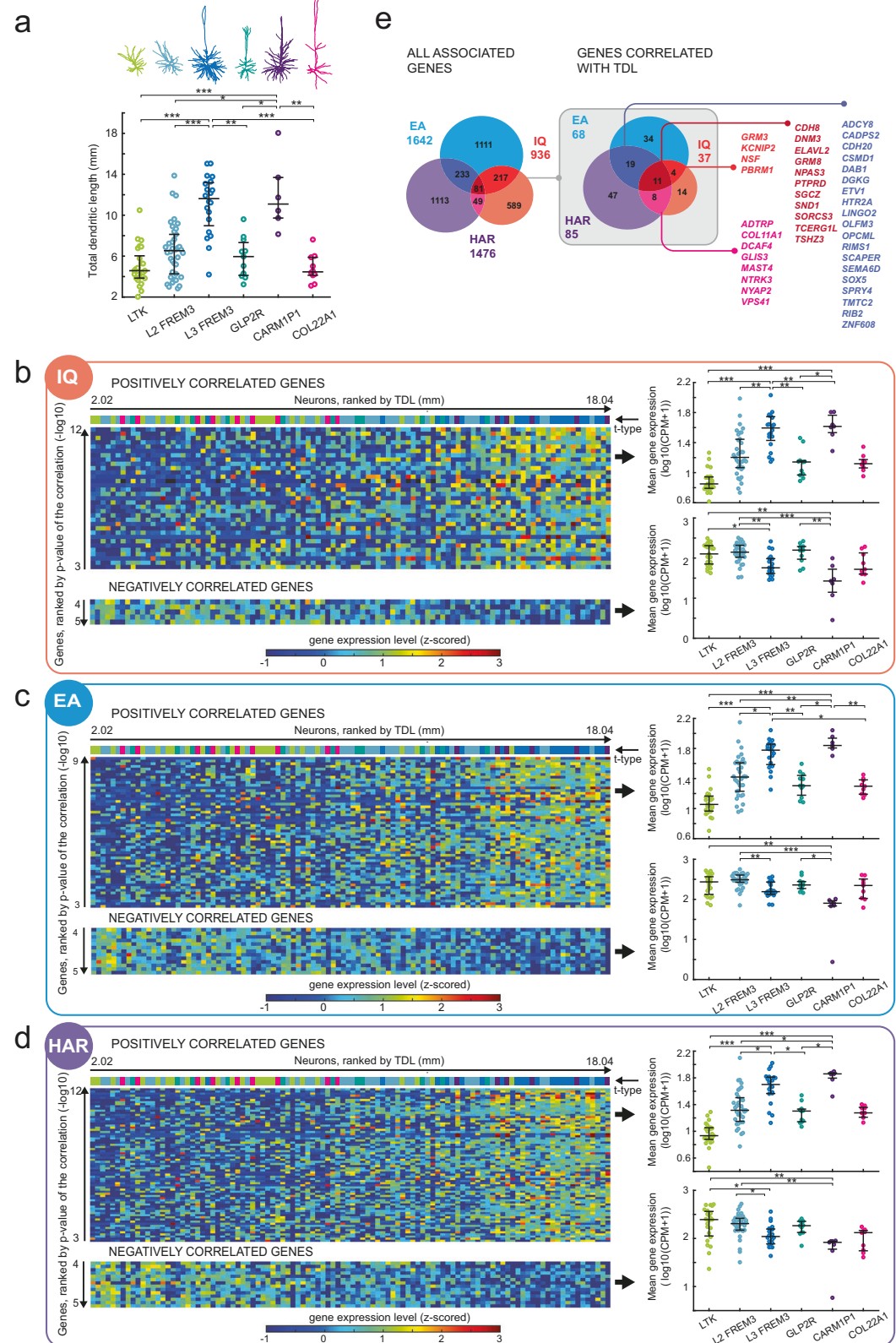

or AP rise speed (33 genes listed in Fig. 4g) and were at the intersection between the IQ, EA, and HAR gene sets—i.e., that belonged to two of the three gene sets. These genes might be particularly important since their expression is not only related to cellular function (AP rise speed) and morphological structure (TDL), but they are also implicated in both human cognition (belong to IQ and/or EA gene sets) and brain evolution (HAR gene set). Since several genes correlated both with TDL

and AP rise speed, the total gene set of interest comprised 62 unique genes (Fig. 5a, listed names of the genes). We investigated the function of this gene set of interest using GeneCards – the human gene database[32,33]. Firstly, all these genes were protein coding and were involved in several brain related disorders (Table S1). Secondly, functional annotations (Table S1) show that these genes are involved in important synaptic processes: synaptic signaling, such as

**Fig. 3 | Differential expression of genes associated with IQ and EA and HAR is related to dendrite size. a** Top: examples of reconstructed dendritic morphologies for each neuron t-type. Bottom: Total Dendrite Length (TDL) of neuron t-types. K-W test; $p = 5.6 \times 10^{-9}$; N (neurons): LTK = 25; L2 FREM3 = 38; L3 FREM3 = 20; GLP2R = 11; CARM1P1 = 6; COL22A1 = 10. Plots in **a**–**d** indicate median (middle line) and 25th and 75th percentiles (whiskers). **b**–**d** Heatmaps show the expression levels (z-scored log10(CPM + 1)) in color code for each neuron (columns) and each gene (rows) per gene set (B: IQ, C: EA, D: HAR). Neurons are displayed in columns ranked from the smallest (left) to the largest (right) TDL, genes are displayed in rows ranked by the significance of their correlation with TDL (unadjusted *p* values). Only significantly correlated genes after 5% FDR correction are shown. The t-types are indicated by color-coded bars above the heatmaps, same color code as in a. Expression of positively correlating genes is shown in top heatmap, expression of negatively correlating genes is shown in bottom heatmap. Mean expression of

these genes averaged per neuron t-type is shown in the plots in the right panel. Top: K-W tests for positively correlating genes: IQ $p = 1.5 \times 10^{-11}$; EA $p = 7.6 \times 10^{-12}$; HAR $p = 3 \times 10^{-12}$. Asterisks represent *p* values of post-hoc comparisons for only CARM1P1 and L3 FREM3 types. Bottom: K-W test for negatively correlating genes: IQ $p = 1.1 \times 10^{-5}$; EA $p = 9.7 \times 10^{-6}$; HAR $p = 3 \times 10^{-5}$. N (neurons) are the same as in **a**. **e** Overview of the subset of genes that correlated positively or negatively with TDL in the different gene sets. The Venn diagrams represent IQ, EA and HAR gene sets and the number of overlapping genes between gene sets for all associated genes (left) and the subset of genes that were significantly correlated with TDL (right, highlighted in gray). The colored numbers outside the Venn diagrams are the total number of genes in each gene set. Individual gene names of genes overlapping between gene sets are listed to the right. N and statistical results are provided as a Source Data file.

metabotropic glutamate receptors (*GRM8* and *GRM3*), potassium voltage-gated channel interacting protein (*KCNIP2*), sodium/potassium transporting ATPase (*NKAIN2*) involved in maintenance of resting membrane potential, and intracellular signaling genes (*PTPRD*). Several genes are involved in (post)transcription regulation (*SND1, TCERG1L, TSHZ3, ELAVL2, NPAS3*).

Surprisingly, among the TDL- and AP-associated genes in the IQ and EA gene sets, HARs were more abundant than expected by chance (Fig. 5a). In the original complete gene sets from GWAS, the overlap between IQ and HAR genes was 14%. In contrast, in the subset of TDL- and AP-correlated genes, 53% of these genes were also part of HAR gene set (Fig. 5a). Similar results were obtained for EA genes (Fig. 5a): the overlap with HARs of the gene set of interest was 47% compared to 19% overlap in the complete EA gene set. This suggests that cellular mechanisms in L3 pyramidal neurons related to human brain evolution and interindividual differences in intelligence might share genetic origin and converge in these neuron types.

We used gene over-representation analysis to gain more insights into the function and underlying biology of this gene set (62 genes) of interest. This analysis tests whether genes associated with a particular biological function, as represented by Gene Ontology (GO) annotations, are statistically over-represented in the gene set of interest compared to background ratios in all genes. We first applied over-representation analysis (using the clusterProfiler R package[34]) to the gene set of interest to determine in which biological pathways (gene ontology groups) these genes are over-represented. Firstly, we used the Molecular Signatures Database (MSigDB) human ontology gene sets C5[35] to read the GO terms of all annotated genes. This provided us with 19413 genes with annotated GO terms. Further, for each GO term, the analysis compared the proportion of genes in the gene set of interest belonging to this GO with background ratios. For example, in our gene set of interest, 9 of total 62 genes were categorized as belonging to the GO term "glutamatergic synapse" (14.5%, Fig. 5b). In contrast, 372 of 19413 total MSigDB genes are annotated to this GO term (1.9%), showing a much lower background ratio. The over-representation analysis statistically compares this difference and answers whether genes in the gene set of interest are significantly over-represented in this GO term. Significant over-representation was determined when the *p* value, the adjusted *p* value and the *q* value (positive False Discover Rate) were lower than 0.05.

We find that a considerable proportion of these genes of interest (Fig. 5a) were over-represented in several biological functions related to synaptic organization and synaptic signaling (Fig. 5b). More specifically, we find that 31 out of the 62 genes of interest are overrepresented in one or multiple GO terms (Fig. 5b). The heatmap shown in Fig. 5c shows correlation coefficients for each overrepresented gene to TDL and AP rise speed in color: most of the over-represented genes correlated positively with TDL and AP and had high correlation coefficients. One of negatively correlated genes was the gene encoding potassium channel interacting protein (*KCNIP2*) that apart from

directly regulating potassium channel function, coordinates the activity of multiple channels to fine-tune neuronal excitability[36].

Apart from the genes coding for metabotropic glutamate receptors and potassium ion channels mentioned above, several overrepresented genes had important functions in synaptic signaling and plasticity (Table S1). We therefore ran an over-representation analysis specifically designed for synapse related genes using a public knowledgebase SynGo[37]. This analysis tests over-representation in known synaptic GO terms against the background of brain expressed genes. From our gene set of interest, SynGo mapped 15 to Biological Processes (Fig. 5d) and 14 genes to a Cellular Component annotation (Fig. 5e). The correlation coefficients to TDL and AP rise speed of these over-represented genes are shown in Fig. 5f.

Specifically, several genes encoding calcium binding proteins emerged from this analysis: *CADPS2* required for the Ca2+-regulated exocytosis of secretory vesicles, and *CACNA2D3* calcium voltage-gated channel subunit. In addition, among over-represented genes were several genes coding for synaptic channels and receptors, such as serotonin receptor (*HTR2A*), calcium voltage-gated channel subunit (*CACNA2D3*), and a protein involved in regulation of synaptic exocytosis and voltage-gated calcium channels (*RIMS1*). Finally, several genes were important constituents of synaptic junctions, such as *STXBP6, PTPRD* and *PTPRT* genes involved in trans-synaptic signaling and *ADCY8* gene (Adenylate Cyclase 8) that is involved in cAMP signaling and affects synaptic plasticity, learning and memory. These results suggest that at the cellular level, dendritic length and AP kinetics to a large extent are under control of the same genes that are involved both in the process of human brain evolution and in processes that might underlie interindividual differences in cognitive function.

## Discussion

Our findings show that genes associated with interindividual differences in cognitive ability (IQ and EA associated genes) and human brain evolution (HAR genes) overlap to a large extent in their expression patterns and biological pathways in cortical areas and adult neurons associated with human cognitive ability. Genes from all 3 gene sets are preferentially expressed in MTG, in L3 glutamatergic neuronal types L3 FREM3 and CARM1P1 that are human-specialized neuron types selectively lost in disorders of cognition[19,20]. These two L3 neuron types stand out because of their large total dendritic length and fast AP initiation kinetics, cellular properties previously linked to IQ scores in human subjects[13,14,29]. We identified a subset of genes from IQ, EA and HAR gene sets based on their correlation of gene expression levels with total dendritic length and AP rise speeds and show that these genes were selectively up- or downregulated in L3 FREM3 and CARM1P1 types. Finally, this subset of genes shared half of their genes between genes associated with cognition (IQ, EA) and HAR genes, and was overrepresented in biological pathways involved in synaptic function and structure.

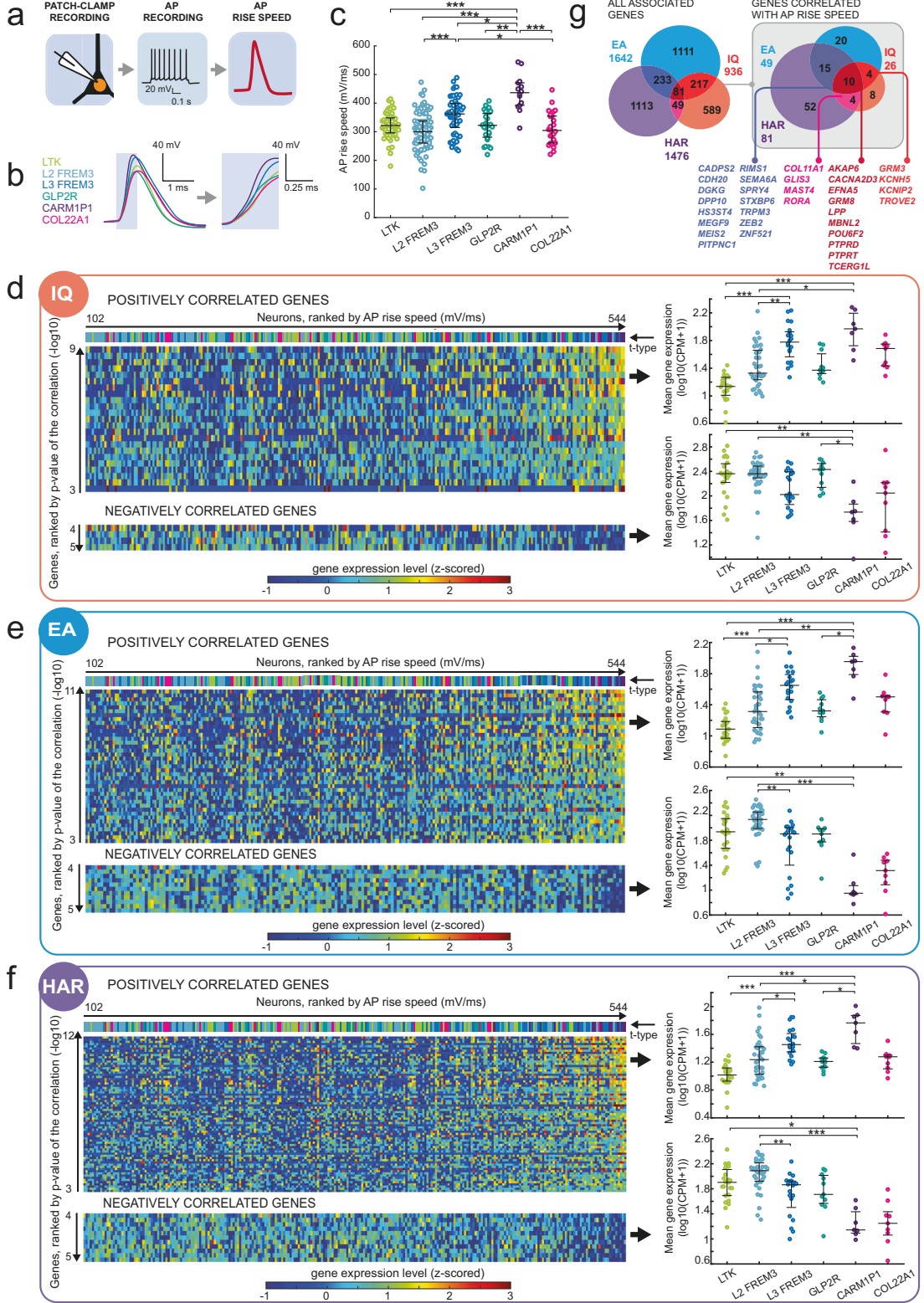

Here we linked findings from GWAS to specific areas, neurons, and cellular phenotypes previously associated with human cognition. GWAS identifies thousands of single nucleotide polymorphisms, only a small proportion of which is directly located in the exonic, protein-coding regions, while the majority are intergenic and intronic. Associated variants that individually have very small effect sizes (<0.1%)[38] are mapped to hundreds of genes that might or might not be

expressed in any tissue and any time point during the development. Here we studied gene expression in adult human neurons originating mainly from non-pathological cortical tissue resected during neurosurgical treatment and in some cases from postmortem brain tissue (snRNA-seq data). Since the neurons come from acutely resected brain tissue, the expression patterns reflect the state of adult human neurons in their intact networks. We find that the majority of

**Fig. 4 | Differential expression of HAR and genes associated with IQ and EA is related to AP rise kinetics. a** Schematic representing the AP recording workflow. **b** Example AP traces of L2/L3 neuron t-types. The highlighted part of the trace represents the rising phase of AP and is shown at higher temporal resolution (right). **c** Mean AP rise speeds for each neuron t-type are shown for the first AP in the recorded trace. K-W test; $p = 2 \times 10^{-9}$; N (neurons) for each group: LTK = 43; L2 FREM3 = 54; L3 FREM3 = 26; GLP2R = 21; CARM1P1 = 12; COL22A1 = 26. **d**–**f** Heatmaps show the mRNA expression levels (z-scored log10(CPM + 1)) in color code for each neuron (columns) and each gene (rows) for the IQ, EA, and HAR gene sets. Neurons are displayed in columns ranked from the slowest (left) to the fastest (right) AP rise speed, genes are displayed in rows ranked by the significance of their correlation with AP rise speed (unadjusted $p$ values). Only significantly correlating genes after 5% FDR correction are shown. The t-types are indicated by color-coded bars above the heatmaps, same color code as in **b**, **c**. Top heatmaps: positively correlating genes. Bottom heatmaps; negatively correlating genes. Right plots: Mean expression of these genes averaged per neuron type. Positively correlating genes: K-W tests: IQ $p = 2.4 \times 10^{-10}$; EA $p = 1.3 \times 10^{-9}$; HAR $p = 4.8 \times 10^{-9}$. Asterisks represent $p$ values of post-hoc comparisons: *$p < 0.05$; **$p < 0.01$; ***$p < 0.001$. Negatively correlating genes: K-W test: IQ $p = 1.3 \times 10^{-4}$; EA $p = 6.5 \times 10^{-9}$; HAR $p = 2.7 \times 10^{-8}$. N (neurons) are the same as in **c**. Plots in **a**–**d** indicate median (middle line) and 25th and 75th percentiles (whiskers). **g** Expression levels of a subset of genes correlated significantly with AP rise speed. The Venn diagrams represent IQ, EA and HAR gene sets and the number of overlapping genes between gene sets for all associated genes (left) and the subset of genes that were significantly correlated with AP rise speed (right, highlighted in gray). The colored numbers outside the Venn diagrams are the total number of genes in each gene set. Complete N and statistical results are provided as a Source Data file.

GWAS-identified genes associated with cognition (92% of IQ genes, 89% of EA genes) and HAR genes (86%) are expressed in these adult neurons and their expression is elevated in specific types. This may suggest that these genes have a role in adult human cognition, and not only during critical developmental periods.

Cognitive functions such as memory, language, and problem-solving require complex integration across numerous cortical regions. The MTG is one of such areas and its important function in cognition has been put forward by several studies of human intelligence that use whole brain morphometric approaches[9–11,39]. Nevertheless, our findings might also apply to other higher-order association areas. Especially superficial layers of the higher-order association cortices are interesting in their relation to cognition and brain evolution[40] These layers show disproportional expansion in the human brain evolution[41], its increased role in higher order association areas[42], and its selective expansion in subjects with higher IQ[13]. However, although outside the scope of this paper, neurons in deeper cortical layers also play a critical role in cortical processing and might present an avenue for future exploration.

Large pyramidal L3 neurons are responsible for cortico-cortical connectivity and integrate many inputs on their vast dendrites. In human subjects, the size of the dendrites in pyramidal neurons directly relates to IQ scores but also to measures of integration within the whole brain network[13,14,29]. The fact that the expression of genes related to human cognition is elevated in these neurons suggests that growth and maintenance of large dendrites and their synapses is at least partially under control of these genes. Indeed, several genes - *CDH8*, *CDH13*, *CDH20*, that significantly correlate with TDL, emerge from over-representation analysis, and overlap across gene sets, - are part of cadherin gene family that are involved in dendrite morphogenesis and synaptic plasticity[43,44]. Furthermore, other genes that emerged from our over-representation analysis have important functions as synaptic receptors (*GRM3*, *GRM8*, *HTR2A*), regulate neuronal excitability (*KCNMA1*, *KCNQ5*, *KCNIP2*, *CACNB4*, *RIMS1*), calcium signaling (*CADPS*, *CADPS2*, *CALB1*), synaptic plasticity (*ADCY8*), and were shown to have neuroprotective effects (*GRM8*)[44]. Moreover, many of the identified genes that correlate with TDL or AP kinetics are also involved in a range of neurodevelopmental disorders (Table S1) marked by changes in cognitive function. These disorders include schizophrenia, autism, epilepsy, depression, mental retardation and intellectual developmental disorder.

Several lines of evidence point out that evolution-driven adaptations of the human cortex are remarkably similar to those that underlie interindividual differences in cognitive ability. Overall brain volume, cortical thickness and thickness of upper cortical layer in higher-order association areas are larger in subjects with higher cognitive performance[17,19] and these areas expanded in human brain evolution. The size and complexity of human dendrites are larger in human neurons[17,19] and associate with higher IQ scores[13,14]. Furthermore, the increased cognitive performance in humans might result from the ability of large human neurons to integrate more inputs and perform faster computations and maintain fast AP speed during high frequency firing[16], and similarly AP speed is faster in individuals with high IQ scores[13,14].

Our results indicate that also at the gene expression level there is evidence of conversion in biological processes involved in both human brain evolution and interindividual differences in cognitive function. A subset of genes that are associated with human cognition and accelerated evolution in humans relate to dendritic size in specific neuronal types and areas of cognition. These genes might serve as promising targets for future therapeutic application in diseases involving cognitive decline.

## Methods

### Gene sets

IQ gene set: IQ gene set is taken from the gene set of 1016 genes reported by Savage and colleagues[5]. These genes were linked to variation in intelligence via positional mapping, expression quantitative trait locus (eQTL) mapping, chromatin interaction mapping, and gene-based association analysis. The gene names are reported in Supplementary Table 12 (859 genes implicated by positional, eQTL, or chromatin interaction mapping of SNPs associated with intelligence) and Supplementary Table 15 (507 genes significantly associated with intelligence in gene-based association tests, GWGAS) of the original paper[5]. After deletion of overlapping genes, the set comprised 1016 unique genes.

EA gene set: For educational attainment gene set, we used the genes from Lee and colleagues[6] listed in Supplementary Table 7 (DEPICT genes) in that study. We selected the 1838 genes prioritized in the paper based on the 5% threshold at the standard false discovery rate (FDR). After excluding missing data (no gene name present), this list was reduced to 1756 genes used in the present study as an EA gene set.

HAR gene set: HAR gene set comprised 1711 HAR genes from ref. [21] reported in ref. [12] and listed in Supplementary Table 2 in that study. These HAR genes were originally presented by Doan and colleagues[21]. Genes that had zero expression across more than 95% of cells in Patch-seq dataset were excluded from the analysis.

### RNA-seq data analysis

For RNA-seq data analysis we used publicly available dataset of RNA-sequenced human brain cells from cortical regions. Data and detailed methods can be found at http://celltypes.brain-map.org. Briefly, for human tissue this method involved layer dissections of human cortical section, neuronal nuclei staining (NeuN) and Fluorescence-activated cells sorting (FACS) isolation, followed by Smart-seq v4 based library preparation and single-cell deep (2.5 million reads/cell) RNA-Seq[18]. For cortical areas analysis we used Allen Cell Types Database -- Human Multiple Cortical Areas – Smart-Seq (2019) and for MTG analysis we used the data from Allen Cell

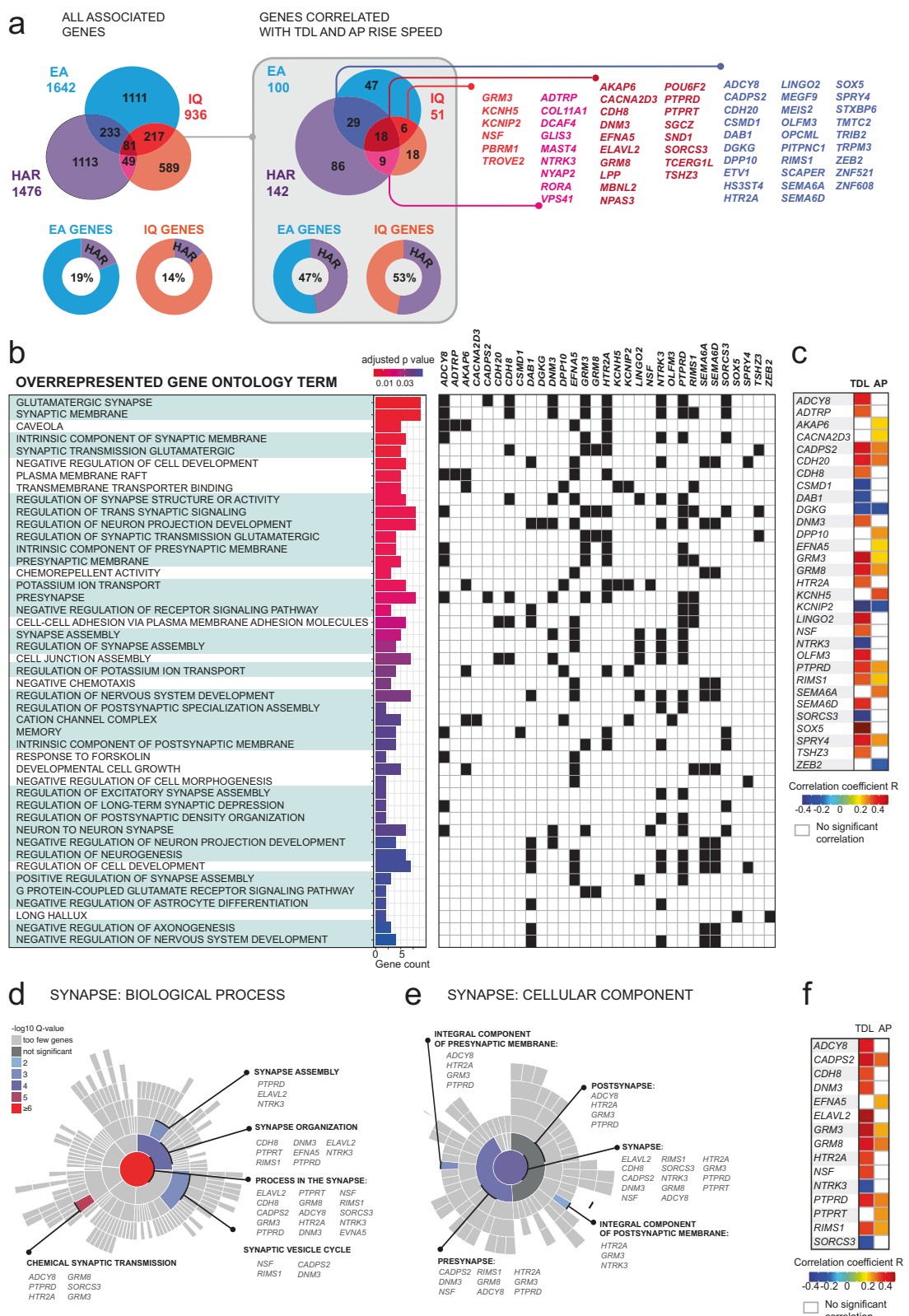

Types Database – MTG – Smart-Seq (2018)[24]. Violin plots in Figs. 1 and 2 represent distribution of mRNA expression (combined exons and introns) that were extracted from these databases and normalized by converting the number of reads into counts per million (CPM + 1) on a log 10 scale, for this conversion total reads of each gene of interest was divided by the total number of reads for all genes in this cell.

**Human tissue acquisition for Patch-seq data collection**

The Patch-seq data were collected and presented in a previous study[19], except for several newly added samples (18 samples) that were collected in this study. All the analyses performed in this study have not been previously published.

Human cortical brain tissue was resected during neurosurgical treatment of epilepsy or tumor in order to gain access to deeper

**Fig. 5 | TDL- and AP-correlated genes are involved in synaptic processes. a** IQ, EA and HAR gene sets and the number of overlapping genes between them are shown for the full sets (left) and for the genes that were significantly correlated with TDL or AP rise speed (right). Lower panels: The percentage of overlapping genes between IQ and HAR and EA and HAR gene sets; one-sided Fisher test, IQ genes: $p = 2.8 \times 10^{-10}$; EA genes: $p = 1.1 \times 10^{-9}$. **b** Over-represented gene ontology (GO) terms in which a selection of genes from the gene set of interest are significantly enriched at 5% FDR. GO terms related to synaptic function and structure are highlighted in green. The bar plot shows the number of genes annotated against each GO term and the significance of overrepresentation in color code ($p$ value adjusted for 5% FDR, one-sided Fisher exact test). The gene names of over-represented genes are shown in the table (right) and marked black if they are overrepresented in each GO term. **c** Over-represented genes from **b** are listed and the correlation coefficients of

their expression levels with TDL and/or AP rise speed are shown in color code. **d** Over-representation analysis for only synaptic GO terms compared to a background set of brain expressed genes. The over-represented Biological Processes in **d** and Cellular Components in **e** are visualized as "sunburst plots." The top-level GO terms "process in the synapse," and "synapse" respectively, are represented by a circle in the center of the sunburst, terms on the second and subsequent hierarchical levels are shown from the center outwards. Each over-represented term is color-coded: the color code shows the $q$ value ($p$ value of one-sided Fisher test adjusted for 1% FDR), the overrepresented genes are listed for each term. **f** Over-represented genes from **d**, **e** are listed and the correlation coefficients of their expression levels with TDL and/or AP rise speed are shown in color code. Source data are provided as a Source Data file.

pathological brain structures, typically hippocampus or amygdala. The tissue specimens were collected by two labs (AIBS or VU Amsterdam) in local hospitals (Harborview Medical Center, Swedish Medical Center and University of Washington Medical Center, Vrije Universiteit Amsterdam Medical Center) in collaboration with local neurosurgeons. All experimental procedures were approved by hospital institute review boards before commencing the study and all subjects provided written informed consent for the use of data and tissue for scientific research. All data were anonymized. The tissue was transported and processed in the labs using the same procedures.

Resected cortical tissue was placed in slicing artificial cerebrospinal fluid (ACSF) immediately following resection. Slicing ACSF comprised (in mM): 92 N-methyl-d-glucamine chloride (NMDG-Cl), 2.5 KCl, 1.2 NaH$_2$PO$_4$, 30 NaHCO$_3$, 20 4-(2-hydroxyethyl)-1-piperazineethanesulfonic acid (HEPES), 25 D-glucose, 2 thiourea, 5 sodium-L-ascorbate, 3 sodium pyruvate, 0.5 CaCl$_2$.4H$_2$O and 10 MgSO$_4$.7H$_2$O. Before use, the solution was equilibrated with 95% O$_2$, 5% CO$_2$ and the pH was adjusted to 7.3 by addition of HCl. Osmolality was verified to be between 295–305 mOsm kg$^{-1}$. Human surgical tissue specimens were immediately transported (15–30 min) from the hospital site to the laboratory for further processing.

In all subjects, the resected neocortical tissue was not part of the epileptic focus or tumor and was removed to access deeper lying structures. The non-pathological status of tissue was based on the surgeon's assessment, pre-surgical MRI scans and the histological assessment of tissue slices after DAPI, DAB, and/or NeuN staining (the integrity of layer structure, absence of abnormal cells or gliosis). We and others refs. [18,19,45,46] have repeatedly demonstrated that using access tissue samples, one can study non-pathological properties of human circuits[13,16,47–51].

### Tissue processing

Human tissue blocks were cut in coronal slices (350 µm thick) with a Compresstome VF-300 (Precisionary Instruments) or VT1200S (Leica Biosystems) vibrating microtome. Brains or tissue blocks were mounted for slicing with the optimal orientation for preserving intactness of apical dendrites of neocortical pyramidal neurons. Slices were transferred to an oxygenated and warmed (34 °C) slicing ACSF for 10 min, then transferred to room temperature holding ACSF of the composition[19] (in mM): 92 NaCl, 2.5 KCl, 1.2 NaH$_2$PO$_4$, 30 NaHCO$_3$, 20 HEPES, 25 D-glucose, 2 thiourea, 5 sodium-L-ascorbate, 3 sodium pyruvate, 2 CaCl$_2$.4H$_2$O and 2 MgSO$_4$.7H$_2$O until transferred for patch clamp recordings. Before use, the solution was equilibrated with 95% O$_2$, 5% CO$_2$ and the pH was adjusted to 7.3 using NaOH. Osmolality was verified to be between 295–305 mOsm kg$^{-1}$.

### Patch clamp recording and AP analysis

Slices were bathed in warm (32–34 °C) recording ACSF containing the following (in mM):[19] 126 NaCl, 2.5 KCl, 1.25 NaH$_2$PO$_4$, 26 NaHCO$_3$, 12.5 D-glucose, 2 CaCl$_2$.4H$_2$O and 2 MgSO$_4$.7H$_2$O (pH 7.3), continuously

bubbled with 95% O$_2$ and 5% CO$_2$. The bath solution contained blockers of fast glutamatergic (1 mM kynurenic acid) and GABAergic synaptic transmission (0.1 mM picrotoxin). Patch pipettes (3–5 MOhms) were filled with ~1.5 µl of internal solution containing biocytin (110 mM potassium gluconate, 10.0 mM HEPES, 0.2 mM ethylene glycol-bis (2-aminoethylether)-N,N,N′,N′-tetra-acetic acid, 4 mM potassium chloride, 0.3 mM guanosine 5′-triphosphate sodium salt hydrate, 10 mM phosphocreatine disodium salt hydrate, 1 mM adenosine 5′-triphosphate magnesium salt, 20 µg ml$^{-1}$ glycogen, 0.5 U µl$^{-1}$ RNAse inhibitor (Takara, 2313 A) and 0.5% biocytin (Sigma B4261), pH 7.3). Recordings were sampled at frequencies of 50 kHz and lowpass filtered at 10 kHz using Multiclamp 700 A/B amplifiers (Axon Instruments). Commands were generated, signals were processed and amplifier metadata were acquired using MIES (https://github.com/AllenInstitute/MIES/), written in Igor Pro (Wavemetrics). Before data collection, all surfaces, equipment and materials were thoroughly cleaned in the following manner: a wipe down with DNA away (Thermo Scientific), RNAse Zap (Sigma-Aldrich) and finally with nuclease-free water.

For human slices, pyramidal shaped neurons in L2-3 were targeted. As described in ref. [19], after formation of a stable gigaseal and break-in, the resting membrane potential of the neuron was recorded. A bias current was injected, either manually or automatically using algorithms within the MIES data acquisition package, for the remainder of the experiment to maintain that initial resting membrane potential. APs were recorded as membrane potential responses in current clamp mode in response to a series of scaled to rheobase depolarizing current steps (of 1 s duration). For the AP data analysis, AP waveforms were extracted from the traces. For each neuron, the mean values of AP rise speeds was determined for the first AP in at each supra-threshold depolarizing current step. AP rise speed was defined as the peak of AP derivative (dV/dt)[13].

Upon completion of electrophysiological examination, a small amount of negative pressure was applied (~−30 mbar) to extract the nucleus. Once the pipette seal reached >1 GΩ and the nucleus was visible on the tip of the pipette, the pipette was rapidly removed and the contents (~1.0–1.5 µl internal solution, cytosol and nucleus) were expelled into a PCR tube containing the lysis buffer (Takara, 634894).

### RNA-seq data processing

The RNA-processing and sequencing were performed as described[19]. In short, collected nuclei from Patch-seq were processed using the SMART-Seq v4 Ultra Low Input RNA Kit (Takara, 634894), cDNA Library Preparation (Illumina FC-131-1096). Each sample was sequenced to approximately 1 million reads. Only uniquely aligned reads were used for gene quantification. Expression levels were calculated as counts of exonic plus intronic reads and log10 (counts per million (CPM) + 1)-transformed values were used. To determine the t-type of each cell, the Patch-seq transcriptomes were mapped to the reference dataset of dissociated human nuclei from MTG[18].

## Morphological reconstruction

A horseradish peroxidase (HRP) enzyme reaction using diaminobenzidine (DAB) as the chromogen was used to visualize the biocytin-filled cells after electrophysiological recording, and 4,6-diamidino-2-phenylindole (DAPI) stain was used to identify cortical layers[19]. Tiled image stacks of individual cells were acquired at higher resolution with a 63 × objective lens (Zeiss Plan-Apochromat 63 × /1.4 Oil or Zeiss LD LCI Plan-Apochromat 63x/1.2 Imm Corr) at an interval of 0.28 μm (1.4 NA objective) or 0.44 μm (1.2 NA objective) along the $z$ axis. Reconstructions of the dendrites were generated for a subset of neurons with good quality transcriptomics, electrophysiology and biocytin fill. Reconstructions were generated based on a 3D image stack that was run through a Vaa3D-based image processing. The total dendritic length was calculated as the sum of total basal and apical dendrites for each cell. Because we had access to the exact location of each neuron within L2/L3, we split FREM3 type based on relative depth of somas within cortical layers: L3 FREM3 neurons were defined as occupying lowest 70% of L2/L3 (relative depth > 0.3), while L2 FREM3 neurons were defined as occupying upper 30% (relative depth < 0.3).

## Statistical analysis

For RNA-seq data analysis, normalized RNA-expression (log10(CPM + 1) values) were averaged across genes within each (IQ, EA and HAR) gene set for individual cells and compared across brain areas, cell classes and t-types pooled for all donors (Fig. 1) and separately within each donor using two-sided Kruskal–Wallis test with Dunn's multiple comparisons test with Holm correction test (Fig. S1). Next, the data originating from one donor was collapsed to a median value and analyzed using Friedman test, followed by Dunn's multiple comparisons test with Holm correction. Analyses were performed using Python based statistical package SciPy 1.0: Fundamental Algorithms for Scientific Computing in Python and Pingouin 0.5.3: statistics in Python. Statistical analyses for Patch-seq data were performed using Kruskal–Wallis tests with Tukey correction for multiple comparisons in Matlab (R2021a, Mathworks).

To determine the individual genes that correlated to TDL or AP speed, linear regression analysis was performed for each gene. Subsequently, genes were ranked based on the significance of the correlation $p$ value. Corrections for multiple testing were performed according to the Benjamini–Hochberg False Discovery Rate (FDR) procedure using FDR of 0.05. During the correction procedure, the critical $p$ values for each gene set were determined; (critical $p$ values for TDL correlations with genes in IQ, EA and HAR gene sets were 0.001; 0.0018 and 0.002, respectively). Only genes with $p$ value lower than critical $p$ value were considered significantly correlated and used for overrepresentation analysis in Fig. 5.

## Overrepresentation analysis and gene ontology

We performed over-representation analysis using clusterProfiler package (v4.4.1)[34] in R an open-source programming environment (4.2.0) and using the SynGo online open-source knowledgebase[37]. Overrepresentation (or enrichment) analysis is a statistical method that determines whether genes from pre-defined sets categorized as belonging to the same gene ontology (GO) group are present more than would be expected (overrepresented) in a subset of data. The analysis estimates the proportion of genes that belong to a certain GO category in a subset of data and statistically compares this proportion to the proportion of all genes in this GO category in the whole genome. For statistical comparisons Fisher's test is used with FDR of 0.05. In our analysis, the subsets of interest were the subsets of genes in IQ, EA or HAR gene sets that significantly correlated to TDL after FDR of 0.05 correction. We used Molecular Signatures Database (MSigDB database v7.5.1, a joint project of UC San Diego and Broad Institute, www.gsea-msigdb.org/gsea/msigdb)[35] to get access to the gene ontologies (ontology gene sets C5) for Homo Sapiens species. To investigate the function of single genes we used GeneCards—the human gene database (www.genecards.org)[32–34].

## Reporting summary

Further information on research design is available in the Nature Portfolio Reporting Summary linked to this article.

## Data availability

The snRNA-sequencing data used for the analysis in Figs. 1, 2 and Fig. S1 were obtained from publicly available database at https://celltypes.brain-map.org/ as datasets (Allen Institute for Brain Science 2020; 2018). Patch-seq data used for the analyses in Figs. 3–5 are publicly available at https://celltypes.brain-map.org/ (Allen Institute for Brain Science 2020). The patch-seq data (CPM values, cell types, TDL and AP) used for the analyses in Figs. 3 and 4 are included in a csv file data_cpm.csv, the file and associated scripts for analysis are provided at the repository DataverseNL: https://doi.org/10.34894/YZFQAO. GO data used for the analysis in Fig. 5 were obtained from MSigDB database available at https://www.gsea-msigdb.org/gsea/msigdb. Overrepresentation analysis of synaptic genes in Fig. 5 was performed using the open-source knowledgebase SynGo available at https://www.syngoportal.org. All data in the figures, N numbers, the results of statistical analyses and post-hoc tests are reported in the Source Data file and provided with this paper. Source data are provided with this paper.

## Code availability

The code generated and used for the data analysis in this study has been deposited in the Github repository[52] under the following https://zenodo.org/badge/latestdoi/565888276.

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

## Acknowledgements

We thank Prof dr. Danielle Posthuma and dr. Christiaan de Leeuw for their insightful comments on earlier drafts of the manuscript and dr. Sophie van der Sluis for the help with the statistical analyses. This study was supported by several grant awards including award U01MH114812 and UM1MH130981-01 from National Institute of Mental Health (NIMH), grant no. 945539 (Human Brain Project SGA3) from the European Union's Horizon 2020 Framework Program for Research and Innovation, the Netherlands Organization for Scientific Research (NWO) Gravitation program BRAINSCAPES: A Roadmap from Neurogenetics to Neurobiology (NWO: 024.004.012). N.A.G. is supported by VI.Vidi.213.014 grant from the Netherlands Organization for Scientific Research (NWO). H.D.M. is supported by ERC AdG 'fasthumanneuron' 101093198.

## Author contributions

S.L.W.D., A.A.G., and D.B.H. performed experimental studies and data analysis, wrote the manuscript, visualized data. I.J.P., R.W., E.J.M., F.W., T.S.H., and L.C. conducted experiments. J.R.M. performed reconstructions of cell morphologies. S.I., P.C.d.W.H., and D.P.N. performed neurosurgery. C.P.J.d.K. supervised the work. B.R.L., K.S., J.T.T., and E.S.L.

contributed Patch-seq data and analysis. H.D.M. supervised the work, wrote and edited the manuscript. N.A.G. performed data analysis, wrote and edited the manuscript, supervised the work.

## Competing interests

The authors declare no competing interests.
