## [Peer Review File · Nature Communications]

Genes associated with cognitive ability and HAR show overlapping expression patterns in human cortical neuron typesREVIEWER COMMENTS

Reviewer #1 (Remarks to the Author):

The authors try to find out if HAR genes and genes associated with human cognition (IQ and EA) are expressed at similar levels in human adult human cortical neurons and certain brain areas (MTG, CgG, A1, M1, V1, S1). They observe that the genes from these 2 gene sets are preferentially expressed in L3 excitatory neurons in the MTG. Moreover, expression levels positively correlate with dendritic tree arborization and firing kinetics. A subset of these genes could be identified to be associated with dendritic length, synaptic function and high abundance of HARs. The resulting hypothesis is that human brain evolution and intelligence associated genes are within the same pool of genes. Genes associated with intelligence are correlated to a cell population in the MTG when comparing the MTG with 4 other brain regions.

Concerns:

- only genes in adult human cortical neurons are investigated. Many or most genes impacting intelligence are developmentally regulated and have peak expression during developmental growth phases such as from late prenatal to young adolescence.
- what are the actual experiments done in this study? RNA-seq is from Allen publicly available, Patch-seq is from earlier work or is this new data? Is everything here re-use of already existing raw data? Unclear.
- It is unclear why in the abstract why larger dendrites are listed as one finding and association with dendritic length is listed another finding. Is “larger dendrites” not the same as dendritic length? Is the point the authors wish to make that a positive correlation of gene expression and neuronal morphology exists, up to the degree that highest correlation contains 50% genes that overlap with HARs?
- In 1D, they control for differences in total expression level amongst the different cortical areas. It would be nice to see this normalization when comparing Glutamatergic, GABA, and Non-Neuronal transcription levels in 1C. Moreover, any explanation why M1 and V1 are so much lower in all three gene sets?
- Fig.2:
 - o The paper mentions L2/L3 neurons being associated with human cognitive function. Because the data is available, it would be more convincing to look at IQ, HAR, and EA transcription levels throughout L1 – L6 in MTG before jumping straight into L2/L3. Although it’s not the focus of the paper, it’s a myopic view of the available data.
 - o If there were other layers that showed overexpression of these genes, it would create an avenue for further exploration. If no other layers show overexpression, the results would be more robust.
 - o if this analysis isn't possible with current cell subtype characterizations, this should be mentioned in the paper
 - o Is there a control for total gene transcription? If not, why?
 - o How does the data in Fig.2 look like when normalized to HK genes?
 - o mean value is indicated instead of median as in Fig.1. It should be briefly explained why. However, in bottom part of figure legend it says again “median”. Unclear.
 - o examples of HRP treated biocytin-filled neurons are missing. The actual microscopic pictures.

- Fig.3: It should be mentioned why here no normalization to HK genes but permutation approach instead.
- Fig.4E: low number of n. Only 3 for CARM1P1 – the most critical neuron type because human specific? This should be explained or more neurons are needed. N of 30 is common. At least as many as for FREM3 should be patched.
- If possible, a short discussion on the insignificant correlation between HAR genes and AP rise speed in figure 4F would be nice.

Reviewer #2 (Remarks to the Author):

See attached Word document.

Reviewer #3 (Remarks to the Author):

It was a pleasure to read this intriguing article where the authors provide evidence that 3 different, but significantly overlapping gene sets (educational attainment, HARs, and IQ) are enriched within brain regions associated with IQ. They follow-up on this finding by investigating the correlation of gene expression patterns from single cell RNA sequencing and Patch-sequencing. Together, these data provide evidence that their overlapping gene-sets are highly correlated with dendritic length. As the authors state, such a correlation helps to understand the potentially shared mechanisms of human brain evolution and the development of human IQ. The current manuscript provides interesting findings with that likely advance the field of human brain evolution.

No Major concerns.

Minor Suggestion:

There has been a lot of interest in the types of genes studied by these authors, but typically prior studies have focused on evolutionary genetics, population genetics (GWAS), or disease associated variants. This study provides a unique assessment of the potential mechanisms by which these genes may contribute to human IQ. The one limitation to the study is that the correlation to IQ associated regions of the brain does not necessarily imply causality. The current study could benefit by adding additional evidence that the regions/genes do cause/contribute to IQ-related features in humans through additional inclusion and discussion of GWAS data and, possibly, more importantly, human disease. It is known that many of the genes they included in their analyses cause a range of neurodevelopmental disorders. It seems that they could strength the case that these genes and the associated dendritic length differences underlie intelligence, which as a term is also somewhat subjective, by including analyses of gene-disease associations. Even more, many of the associated disorders are well-studied in both humans and animal models. Therefore, it is possible that a more conclusive link to specific phenotypes like dendritic length and IQ could be ascertained from the inclusion of data from prior disease studies.

We thank all the reviewers for their thoughtful and detailed comments that helped us greatly to improve the quality of the manuscript. We have thoroughly revised the manuscript following the reviewers' suggestions. In summary, we added 18 extra samples, including 4 rare human-specific CARM1P1 types to increase the sample size. We re-ran, focused and added additional analyses (heatmaps displaying correlations with TDL and AP rise speed with all genes; SynGo over-representation analysis) of this increased dataset. Furthermore, we provide more explanation on the points suggested by the reviewers. We find that all our conclusions were strengthened by this additional revision.

Please find below our point-by-point replies to the reviewers' comments.

REVIEWER COMMENTS

Reviewer #1

Reviewer #1 (Remarks to the Author):

The authors try to find out if HAR genes and genes associated with human cognition (IQ and EA) are expressed at similar levels in human adult human cortical neurons and certain brain areas (MTG, CgG, A1, M1, V1, S1). They observe that the genes from these 2 gene sets are preferentially expressed in L3 excitatory neurons in the MTG. Moreover, expression levels positively correlate with dendritic tree arborization and firing kinetics. A subset of these genes could be identified to be associated with dendritic length, synaptic function and high abundance of HARs. The resulting hypothesis is that human brain evolution and intelligence associated genes are within the same pool of genes. Genes associated with intelligence are correlated to a cell population in the MTG when comparing the MTG with 4 other brain regions.

Concerns:

1. Only genes in adult human cortical neurons are investigated. Many or most genes impacting intelligence are developmentally regulated and have peak expression during developmental growth phases such as from late prenatal to young adolescence.

Reply: We agree with the reviewer that many of the genes impacting intelligence are developmentally regulated and their developmental profiles would be extremely interesting to consider. However, due to the absence of Patch-Seq data from adolescent and child donors it is not possible to address this. Nevertheless, the fact that adult neurons express mRNA of these genes suggests that the genes are also functional at the adult stage, contributing to neuronal function and potentially to cognition.

2. what are the actual experiments done in this study? RNA-seq is from Allen publicly available, Patch-Seq is from earlier work or is this new data? Is everything here re-use of already existing raw data? Unclear.

Reply: Single cell RNA-seq data were analyzed from the publicly available Allen Brain database. The Patch-Seq data were collected by us in collaboration with Allen Institute and most of the data were already described in a previous study (Berg et al. 2021) except for several samples (18 new samples) added during the revision. All the analyses performed in this study are novel and not previously published. We clarify this now in the methods section (p 19 lines 23-25).

3. It is unclear why in the abstract why larger dendrites are listed as one finding and association with dendritic length is listed another finding. Is "larger dendrites" not the same as dendritic length?

Reply: Indeed, "larger dendrites" is the same as "larger total dendritic length". We replaced "dendrites" with "total dendritic length" in the abstract to clarify this.

4. Is the point the authors wish to make that a positive correlation of gene expression and neuronal morphology exists, up to the degree that highest correlation contains 50% genes that overlap with HARs?

Reply: Our finding is that when we consider the subset of genes from IQ and EA gene sets that correlate with cellular morphology and physiology, these genes are also for 50% part of the HAR gene set. We clarify this in the text of the Key findings (p 1 line 29) and in Figure 5A.

5. In 1D, they control for differences in total expression level amongst the different cortical areas. It would be

nice to see this normalization when comparing Glutamatergic, GABA, and Non-Neuronal transcription levels in 1C. Moreover, any explanation why M1 and V1 are so much lower in all three gene sets?

Reply: We reconsidered this type of normalization to housekeeping genes and decided to remove this normalization from the manuscript based on the following reasons:

- Firstly, all gene expression levels are already normalized to the number of reads in each sample (cell nucleus). The log₁₀(cpm+1) values for gene expression levels that we used throughout our study are based on count per million reads (CPM). It is obtained by dividing the number of reads mapped for a gene of interest divided by the total number of mapped reads for all genes in this cell and multiplied by a million. Thus, additional normalization is not necessary.
- Secondly, the housekeeping genes are not the best choice for normalization because many of the genes from IQ, EA or HAR gene set are also part of this gene set.

We now clarify this in the manuscript text (p 3, lines 27-32).

6. Fig.2:

6.1. The paper mentions L2/L3 neurons being associated with human cognitive function. Because the data is available, it would be more convincing to look at IQ, HAR, and EA transcription levels throughout L1 – L6 in MTG before jumping straight into L2/L3. Although it's not the focus of the paper, it's a myopic view of the available data. If there were other layers that showed overexpression of these genes, it would create an avenue for further exploration. If no other layers show overexpression, the results would be more robust. If this analysis isn't possible with current cell subtype characterizations, this should be mentioned in the paper.

Reply: We agree that neurons in other cortical layers (especially layers 5 and 6) are also important for cognitive function. Our focus on superficial layers is explained by the disproportional expansion of L2/3 in human brain evolution (Hutsler, Lee, and Porter 2005), its increased role in higher order association areas (Wagstyl et al. 2020), and its selective expansion in subjects with higher IQ (Goriounova et al. 2018; Heyer et al. 2022). However, we acknowledge the important role of these other layers in cortical processing.

We performed additional analysis that shows RNA-sequencing data for glutamatergic t-types per layer (Rebuttal Figure 1). Layer 1 is not included since it does not contain glutamatergic types. Patch-Seq data is not available for cell types in other layers and could not be included. Although outside the scope of this paper, the data in Rebuttal Figure 1 suggest that neurons in deeper cortical layers might present an interesting avenue for future exploration. We added text to discussion mentioning the other cortical layers and explaining our choice of superficial layers (p 18, lines 2-9).

Rebuttal figure 1. Gene expression per layer in MTG. Violins represent distribution of gene expression of all the single cells from all donors together, single data points are median values of expression per donor. Friedman test: IQ: $F=8.4$, $p=0.063$, EA: $F=12.6$, $p=0.0022$; HAR: $F=11.4$, $p=0.0071$

6.2. Is there a control for total gene transcription? If not, why?

Reply: Yes, there is, see the explanation above at point 5, all data are presented as count per million reads (cpm) per cell (nucleus).

6.3. How does the data in Fig.2 look like when normalized to HK genes?

Reply: As we explain above in point 5, we excluded this type of additional normalization from the analysis.

6.4. mean value is indicated instead of median as in Fig.1. It should be briefly explained why. However, in bottom part of figure legend it says again “median”. Unclear.

Reply: ‘mean gene expression level’ refers to the averaged expression levels across genes within a gene set (IQ, EA or HAR) for every cell. The violin plots in Fig 1 represent the data distribution of all cells and the median of this distribution. This holds true for both figure 1 and figure 2. We now clarify this in the text (p 3, lines 25-35).

6.5. examples of HRP treated biocytin-filled neurons are missing. The actual microscopic pictures.

Reply: In this study, we used earlier reconstructed, quality checked and publicly released morphological data (Berg et al. 2021) without using the actual microscopic pictures. However, we provide below in Rebuttal Figure 2 the microscopic images and their digital reconstructions of the examples shown in the manuscript Fig 2A:

Rebuttal Figure 2: microscopic images and their digital reconstructions of the examples shown in the manuscript Fig 2A.

7. Fig.3: It should be mentioned why here no normalization to HK genes but permutation approach instead.

Reply: See explanation above at point 5, the mRNA expression data is already normalized as count per million reads (cpm) per cell (nucleus).

8. Fig.4E: low number of n. Only 3 for CARM1P1 – the most critical neuron type because human specific? This should be explained or more neurons are needed. N of 30 is common. At least as many as for FREM3 should be patched.

Reply: High quality Patch-Seq data from human neurons in surgically resected brain tissue is extremely difficult to obtain, especially for the rare cell types such as CARM1P1. However, we agree with the reviewer that a higher sample number would help to strengthen the conclusions of the paper. Hence, we collected 18 additional Patch-Seq samples, 4 of which are CARM1P1 types, and added these to the analysis.

9. If possible, a short discussion on the insignificant correlation between HAR genes and AP rise speed in figure 4F would be nice.

Reply: we replaced the analysis previously shown in Figure 4 with a more inclusive analysis. Instead of using the mean expression of all genes in each of the IQ, EA and HAR gene sets in CARM1P1 cell types, we now show the data for AP correlated genes for each gene in each cell type in heatmaps in Figure 4. In this analysis, 81 genes from the HAR gene set significantly correlated with AP rise speed: 71 genes showed positive correlation of their expression levels with AP rise speed, while 10 genes showed a negative correlation.

Reviewer #2:

The authors took three sets of genes—“intelligence,” “education,” and “human-accelerated regions.” They found that each set of genes was more strongly expressed in glutamatergic neurons than in GABAergic neurons; their expression in the latter was in turn stronger than their expression in non-neuronal cells (presumably glial). The elevated expression in glutamatergic cells was more pronounced in two cell types, FREM and CARM1P1. These types are characterized by longer dendritic length and action-potential rise speed; the last property could be assessed because the neurons in this analysis were obtained from neurosurgery patients.

I was fairly enthusiastic about this manuscript when I first read through it. As I wrote down my detailed review, however, I came to be somewhat skeptical. I am requesting some revisions and clarifications. These will have to be very thorough and convincing if I am to ultimately sign off on this paper.

Please note that I am not a neuroscientist and trust that at least one reviewer has expertise to evaluate the neurophysiology and related aspects of the paper.

A postdoc, two PhD students, and two research assistants aided me in the review of this manuscript.

GENERAL POINTS

1. Strength of evidence from prior studies. The authors say that “several morpho-electrical properties of human pyramidal neurons in cortical layers 2 and 3 (L2/L3) from higher order association area (middle temporal gyrus, MTG) were shown to associate with intelligence in humans” and cite references 13 and 14. Although very impressive in many ways, the first of these two studies observed 46 subjects and the second 58. Consequently, neither study was able to obtain very strong evidence for most of its claims, by the standard of Benjamin et al. (2018). Therefore I would prefer any reference to this prior work to be softened. For example, the sentence just quoted could be revised to say “several morpho-electrical properties of pyramidal neurons in cortical layers 2 and 3 (L2/L3) from higher-order association area (middle temporal gyrus, MTG) were suggestively implicated as correlates of human intelligence.” A statement like “AP rise speed of pyramidal neurons remains fast during repetitive firing and shows a robust correlation with IQ scores, explaining 58% of variance in IQ” is way out of bounds.

Reply: As requested by the reviewer, we have adjusted the text, toned-down and softened the references to previous work (p2, lines 24-26). We do wish to emphasize that our previously published findings on correlations between properties of neurons in MTG and IQ scores of the same patients were based on biologically-relevant hypotheses that are well-supported by a large body of literature on identified neurobiological substrates of intelligence. Firstly, the neurons are located in a brain region, MTG, that has been structurally and functionally associated with IQ scores in thousands of healthy subjects, reported by multiple independent research groups (reviewed in Deary et al. 2010). Secondly, the relation between cortical thickness, neuron density, size, complexity, and synapse numbers within the human brain is well-established (DeFelipe 2011). The association between human neuron properties and IQ scores that we reported in previous work (Goriounova et al., 2018; Heyer et al., 2022) is thereby not a finding that ‘stands on its own’ or resulted from open screens, but is a logical next question based on well-founded structural and functional associations reported in the literature. See also our reply to point 8 below.

2. Definition of gene sets. It is never explained, as far as I can see, where the educational attainment (EA), IQ, and human-accelerated-region (HAR) genes are taken from. E.g., are the EA genes the DEPICT-prioritized genes from Lee et al. (2018)? The MAGMA-prioritized genes? The intersection? Are they defined in some other way? And so on for the other two gene sets.

Reply: We provide more detailed information now on the definition of the gene sets in the text of the paper (in the methods section, p18 line 41 – page 19 line 9).

IQ gene set: The IQ gene set is taken from the gene set of 1,016 genes reported in Savage et al paper (Savage et al. 2018). These genes were linked to variation in intelligence via positional mapping, expression quantitative

trait locus (eQTL) mapping, chromatin interaction mapping, and gene-based association analysis. The gene names are reported in the Supplementary Table 12 (859 genes implicated by positional, eQTL, or chromatin interaction mapping of SNPs associated with intelligence) and Supplementary Table 15 (507 genes significantly associated with intelligence in gene-based association tests, GWAS). After deletion of overlapping genes, the set comprised 1,016 unique genes.

EA gene set: For the educational attainment gene set we used the genes from Lee and colleagues (Lee et al. 2018) listed in the Supplementary Table 7 (DEPICT genes). We selected the 1,838 genes prioritized in the paper based on the 5 % threshold at the standard false discovery rate (FDR). After excluding missing data (no gene name present), this list was reduced to 1,756 genes used in the present study as an EA gene set.

HAR gene set: HAR gene set comprised 1711 HAR genes reported in Wei et al (Wei et al. 2019) and listed in Supplementary Table 2. These HAR genes were originally presented by Doan and colleagues (Doan et al. 2016).

The full lists of all individual gene names in IQ, EA and HAR sets used in this study are reported in the Source Data file.

Incidentally, Lee et al. (2018) is accidentally cited twice; it is both reference 6 and 21.

Reply: We corrected the references.

3. Statistical significance. In a plot like the ones in Figure 1 and thereafter, is it possible for more than one data point contributing to a violin to come from the same individual donor? Or for the same individual donor to contribute data points to more than one violin? Much more detail about such matters is necessary to evaluate whether the statistical testing properly took account of possible non-independence.

For example, if a single person can donate multiple neurons, then one option is to aggregate the data in such to remove any influence of non-independence. The online supplements of Okbay et al. (2016) and Lee et al. (2018) describe some ways to do this with the BrainSpan Developmental Transcriptome, a dataset with a nested structure of this kind. (Some would advise using linear mixed models, but I don't believe that such models can be counted on to deal with non-independence.) Another thing to try might be a "double bootstrap," in which individuals are resampled with replacement and then neurons are resampled with replacement from each individual (with appropriate stratification by cell type) to create a bootstrap replicate.

If a single person can donate neurons of different types, neurotransmitters, or areas that are being compared, then the Wilcoxon signed-rank test might be in order.

Dealing with non-dependence can have a substantial effect on the results. Even though the differences between cell types look very stable in the figures, it is very difficult to judge the numerical soundness of the findings without more information and possibly reanalysis. (Incidentally, I suggest trying to get better resolution of very small P values. E.g., why does the caption of Figure 2 give $4.6E-24$ as one Patch-Seq P value but then impose a floor of $1E-16$ for the RNA-seq P values? One would think that the Patch-Seq P values are less significant because of the reduced sample sizes.)

Reply: We agree with the reviewer and addressed the nested nature of the data.

The high-throughput, single cell RNA-sequencing data were collected from thousands of cells but only from a few human donors (N=3 for all cortical areas, N=4 for MTG). The low-throughput Patch-Seq data with complete physiological, morphological and transcriptomic profile of each cell was collected from more donors (N=57), but with lower numbers of cells per donor. To account for the nested nature of the data, apart from the analysis of all cellular data (violins in Fig. 1), we performed two additional analyses. In the first analysis, we averaged the mean expression of gene sets in cell types per individual donor and compared those across brain areas with donor as N number (data points in Fig 1). In the second analysis, we tested differences in gene expression levels within individual donors, comparing brain areas and cell types with cells as N number (violins in Fig S1). All analyses showed similar results: the expression levels of genes from IQ, EA and HAR gene-sets were lower in primary sensory areas (S1, V1, A1), and higher in MTG, CgG and M1 in glutamatergic neuron types (Fig. 1, Fig S1).

Each donor showed a similar pattern of expression levels of genes from the IQ, EA and HAR gene sets across brain areas and cell types. The analyses show that there is not a single donor that stands out, and that variation of gene expression of gene sets between cells within single donors is substantially larger than variation between donors, indicating that the analyses were not biased by nested data. To confirm this, we calculated variance between and within donors and show that variance within is 100-fold larger than between donors (reported in the figure legend of Figure S1: IQ gene set: variance within donors: donor1=0.024, donor2=0.027, donor3=0.025; variance between donors=0.00025. EA gene set: variance within donors: donor1=0.045, donor2=0.053, donor3=0.048; variance between donors=0.00054. HAR gene set: variance within donors: donor1=0.029, donor2=0.033, donor3=0.03; variance between donors=0.00047).

Similar results were obtained using these approaches on the Patch-Seq data (cellular data shown in Fig 2 and donor data in Fig S2), confirming that our original conclusions hold: MTG glutamatergic cell types in general, and layer 3 human-specific types in particular show the highest expression of genes related to human cognition and brain evolution.

The results of these analyses are shown in Figure 1, 2, S1, S2 and explained in the text on p 3 line 41 – page 4 line 6 and p 7 lines 13-28.

We report now the actual *p* values though out the paper, some of these values are extremely small.

4. More on statistical significance. There are serious problems with the statistics summarized in Figure 3. The authors state in the text that “TDL [total dendritic length] did not correlate significantly to the gene expression levels within t-types, possibly because of smaller TDL variability within types than between types.” I am not sure that I buy the explanation, because in Figure 3A some of the types do seem to contain substantial variability in TDL. It may be that the expression levels of the genes in these sets does not have a casual effect on TDL, or has only an indirect effect through specifying the cell type and no direct effect beyond that, or ... Whatever the explanation, the fact that within-type variation in expression is uncorrelated with TDL poses a problem for the statistical tests. The *P* values in the panels B-D are not meaningful because the key assumption for the validity of the *t* test in univariate linear regression is that, conditional on the values of the independent variable, the deviations of the data points from the true regression line are independent. But that assumption is violated in this dataset; if two data points have the same mean gene expression but come from the same cell type, their residuals will be correlated. Fitting regression lines to the scatter plots is highly misleading in this case anyway, precisely because there does not seem to be any relationship between the independent and dependent variables within cell types. Finally, the printing of regression *P* values in panels B-D is highly confusing because the permutation *P* values implied by the histograms beneath the scatter plots are not consistent with them.

I think that the confidence bands and printed *P* values in panels B-D must be removed.

It is also possible that the permutation test can be improved. As I wrote above, it is not clear where these gene sets come from. But there is a tendency in GWAS for prioritized genes to be longer, regardless of the prioritization method. (E.g., if a locus is constructed around a lead SNP, longer genes are inherently more likely to overlap that locus.) Brain-expressed genes also tend to be longer, leading possibly to spurious findings. If the prioritization method favors longer genes (which is true of DEPICT, nearest gene, all genes in loci), it would be a good idea to choose permutation gene sets that come within some tolerance around the actual gene set’s total length.

Reply: We thank the reviewer for this insightful comment and reconsidered our approach. In fact, we removed the linear regression analysis on the mean gene expression per gene set completely. We agree that the modest *n* numbers per pyramidal neuron t-type precludes strong conclusions based on correlations within a single type. The previously used **mean** expression levels of a given gene set in a cell did not account for the fact that not all genes within a gene set might associate with cellular electrical function and structure. Rather, some of these genes may be more important than others and only these genes may have increased or decreased expression in neurons with larger TDL or faster APs. Therefore, we now analyzed correlations with TDL and AP rise speed for **each** gene within the IQ, EA and HAR gene sets separately in all pyramidal neurons. We z-scored mRNA expression of each gene and plotted it against TDL in Figure 3 and AP rise speed in Fig 4. This allows us to find the potentially more relevant genes based on their correlation strength (*p*-value in linear regression) and significance after 5 % FDR correction. Indeed, many genes from the gene sets did not show significant correlations of expression with the neuronal properties, but a subset of genes of each of the gene sets showed

strong positive or strong negative correlations of expression with TDL (summarized in fig 3E) or AP rise speed (summarized in fig 4G). We confirmed that for those significantly correlated genes there was also a significant mean difference in expression levels across cell types (scatter plots on the right in Figs 3B-D and Figs 4D-F). Finally, as an example, we show the relationship between mean gene expression and t-types for the set of TDL-positively correlating IQ genes (from Figure 3B) in Rebuttal Figure 3. The regression analysis was significant also within t-types for those t-types that had a sufficient sample size (LTK, L2 FREM3, L3 FREM3, GLP2R). The Durbin-Watson test for independence of residuals was not significant ($p=0.48$) confirming that residuals were independent.

Rebuttal Figure 3: Relationship between total dendritic length (TDL) and mean gene expression for TDL-correlating IQ genes, per cell t-type.

Also, since the regression R -squared is not a good summary statistic, a fairer comparison might be based on the mean difference in expression between the FREM/CARM1P1 cell types and the others.

Reply: We now included the mean difference in expression between cell types for all correlating genes in Figures 3B,C,D and 4D,E,F.

Many of the same comments apply to Figure 4. The confidence band and printed P value in panel E should be removed. Perhaps these can be retained in panel F, subject to the issue of non-independence being properly addressed (see above). Perhaps most importantly, it should be emphasized that the evidence for levels of gene expression being associated with action-potential rise speed is not particularly strong (Benjamin et al., 2018); with this small a sample size (8 neurons), it could not be otherwise. For example, instead of “linear regression analysis showed that expression levels of IQ and EA genes robustly correlated with AP rise speeds in CARM1P1 types” ($p.9$), it should be “linear regression analysis tentatively showed that expression levels of IQ and EA genes are correlated with AP rise speeds in CARM1P1 types.”

Reply: Similar to the analysis of looking at expression of the individual genes from the different gene sets in relation to total dendritic length, we have now analyzed the expression of the individual genes from the different gene sets in relation to AP rise speed, now shown in Figure 4. Many genes do not show correlation of expression with AP rise speed, but a small number do show either positive or negative correlation (after false-positive correction) with AP rise speed, summarized in Fig 4G.

Here is a more substantive question about non-independence. Suppose that differences in gene expression between cells *from the same individual* contribute to some correlation. (Maybe this is not the case. In any event, the paper needs to make this information more accessible.) For example, suppose that even within the same individual, cells expressing EA/IQ/HAR genes at a higher level show a greater total dendritic length. How would the authors interpret such a trend in a paper aimed mainly at explaining individual differences (i.e., differences between people)?

Reply: See our reply above to the Reviewer's comment in point 3. The aim of the present study is to test whether expression of genes associated with IQ and EA, and HAR genes are correlated with brain areas and cell types that have been associated with IQ, i.e. MTG (Deary, Penke, and Johnson 2010) and glutamatergic cells in L2 and L3 of MTG (Goriounova et al. 2018; Heyer et al. 2022), and identify those genes. We show that within single donors this is consistently the case. The GWAS studies from which the IQ and EA gene sets were obtained already showed that these genes can explain part of the individual differences between people in IQ scores or EA, as do the MRI studies on thousands of subjects showing that structure and function of MTG can explain part of the individual differences between people in IQ scores, as did our previous work on MTG micro-architecture showing that MTG layers and pyramidal neuron properties in MTG can explain part of the individual differences between people in IQ scores.

5. Enrichment (over-representation) analysis. This part of the paper is extremely difficult to understand. The authors state that "over-representation analysis determines whether genes that belong to the same biological process are present more than would be expected (over-represented) in the selected gene subsets: TDL-correlated subsets of IQ, EA and HAR genes." So far, so good. But the next sentence is mystifying: "We find that large proportion of these genes emerged from the over-representation analysis (5/23, 22% in IQ gene set; 24/64, 37% in EA gene set, and 22/66, 33% in HAR gene set) and the biological processes in which they were over-represented almost exclusively related to synaptic organization and synaptic signaling (Fig 5D)." What does this mean? How does a gene qualify to be counted in the numerator of a fraction like 5/23, 24/64, 22/66? I suppose one possibility is that a gene makes it if it is a member of at least one MSigDB gene set emerging as statistically significant in the over-representation analysis, but if so this needs to be clearly explained.

I don't think it is anywhere stated what qualifies a MSigDB gene set to be statistically significant in the over-representation analysis. Is it just $P < 0.05$, or $FDR < 0.05$, or Bonferroni significant, or what?

Reply: In the revised manuscript, we focus the over-representation analysis on the subset of 62 IQ, EA and HAR genes that significantly correlated with TDL or AP rise speed and that belonged to at least two gene sets (Figure 5A). These genes might be especially important since their expression is not only correlated to cellular function (AP rise speed) and structure (TDL) but they have also been implicated both in human cognition (IQ and EA) and brain evolution (HARs). Furthermore, we ran this analysis using an additional tool – SynGo - that focuses on exclusively synaptic genes (and proteins). We now better clarify the over-representation analysis in the manuscript text on p 14 line 44 – p 15 line 13 and lines 21-27.

6. Focus on medial temporal gyrus. My reading is that medial temporal gyrus (MTG) is often a brain region of interest, but only because of limitations in the data resource (Allen Cell Types Database) or the manner in which neurosurgery is often conducted. If this is really the case, then I think some effort should be made to avoid playing up the role of the MTG in cognition. The findings about gene expression, DTL, and so forth may hold even in other brain regions (e.g., prefrontal cortex).

Reply: Although MTG is a prominent source of surgically resected tissue, its important function in cognition and IQ has been shown by many studies of human intelligence, both its structure as well as its function measured in structural and functional MRI in thousands of subjects (Deary, Penke, and Johnson 2010; Choi et al. 2008; Colom et al. 2009; Karama et al. 2009). Nevertheless, we agree that our findings might also apply to other higher-order association areas and mention it in the discussion (p 18 lines 2-4).

7. The meaning of axon-potential rise speed. I don't believe that the authors ever get across a clear explanation or speculation of how AP rise speed might affect information processing. The paper might be strengthened by something along these lines, although the authors should not go *too* far in this direction (since the statistical evidence for a link between AP rise speed and intelligence is arguably not that strong).

I do want to provide some counterarguments against a naïve explanation. It is tempting to think that AP rise speed might be beneficial because it speeds up the propagation of an impulse along an axon. If a recording location reaches its peak voltage sooner, then the "next location" will do so as well, and so on. But there are some reasons to think that the modest difference implied by Figure 4B cannot be very consequential.

Nerve conduction velocity depends on the precise part of the precise nerve assayed, but it shows a population mean of typically 50-60 meters/second with a standard deviation 10-20 times smaller than the mean (Reed & Jensen, 1991; Stetson et al., 1992). If we generously stipulate that the distance between the occipital lobe (the

site of visual processing) and the frontal lobe (the site of integration and decision-making) is 20 centimeters, then the axonal contribution to the relaying of a message from occipital to frontal lobes takes about 5 milliseconds—a rather minute proportion of the roughly 200 milliseconds required by a human to perform even the simplest speeded cognitive task, no matter where the person's conduction velocity lies within the range of individual differences. The most distant neurons in the brain are probably separated by at least half a dozen synapses (Sporns, 2011), and thus it may well be the case that communication between brain regions is delayed more by synaptic transmissions (each taking about 1 millisecond) than by axonal propagation. How can it be, then, that the precise speed of action potentials has a substantial impact on human intelligence?

Here is another point weighing against the importance of axonal conduction velocity. Previous studies have found that SNPs lying in or near genes highly expressed in oligodendrocytes (which wrap axons in myelin and thus speed up action potentials) *do not* account for an enriched share of the heritability of educational attainment (Finucane et al., 2018; Lee et al., 2018; Okbay et al., 2022). Savage et al. (2018) found something similar for IQ, although their analysis may not have been statistically powerful. The current paper indeed supports these earlier findings, in that its gene sets are expressed at higher levels in glutamatergic and GABAergic neurons than in non-neuronal (presumably glial) cells (Figure 1). (Incidentally, it may be politic to cite these earlier papers in this connection.)

Reply: We wish to apologize to the reviewer that our explanation of the relevance of AP rise speed for cognition created confusion. The rise speed of AP and the advantage it provides for neuronal computation are **not related to the conduction velocity** in the axon. Rather, AP rise speed, or AP initiation speed, is crucial to neurons to encode information in AP timing, i.e. it allows neurons to react fast to rapidly changing subthreshold membrane potential changes induced by synaptic inputs and convert these inputs into AP output (Goriounova et al. 2018; Testa-Silva et al. 2014; Eyal et al. 2014; Ilin et al. 2013). The faster AP initiation is, the larger the information content (bandwidth) is that can be encoded in AP output timing. Fast AP initiation kinetics (rise speed) is the critical requirement for large bandwidth information processing by neurons and in neuronal networks (Ilin et al. 2013; Volgushev 2016). With a sluggish AP initiation, a neuron will not be able to encode the temporal details present in the synaptic subthreshold membrane potential changes. Human neurons with large dendrites receive many more synaptic inputs than human neurons with small dendrites (DeFelipe, Alonso-Nanclares, and Arellano 2002; DeFelipe 2011), leading to a larger information content (bandwidth) in subthreshold synaptic membrane potential changes. Fast AP initiation kinetics are thereby particularly important for large pyramidal neurons to be able to encode a larger information space into AP output timing. Large neurons with fast AP initiation mechanisms can process and relay much more information. This is relevant for cognition, since neurons are the basic units of cortical information processing. Cortical networks that contain larger neurons that receive more synapses and that have faster AP initiation kinetics can process more information than networks with smaller neurons with slow AP initiation kinetics. We clarify this point in the manuscript text (p 11 lines 20-35).

8. Neuron size, dendrite size, and synapse number. It seems critical to this paper that there exist a relationship between dendrite size and IQ within humans. The authors make much of the fact that humans have larger dendrites than other mammals. I have some questions about these relationships. First, neuron size tends to increase across non-primate species with body size (Herculano-Houzel, 2016). Could it be that dendrites in humans are larger than those in mice simply because humans are bigger overall? Second, the authors say that dendrite size is important because bigger dendrites can form more synapses. But if that is the case, what do the authors make of this passage:

Convergence can be quantified as the number of synapses per neuron. This number has variously been reported to be similar across matching regions of the neocortex or to scale up with brain size (Cragg, 1967; Beaulieu and Colonnier, 1985; DeFelipe et al., 2002). This number has been reported to be 0.6-fold greater in the monkey primary visual cortex compared with the cat (beaulieu and Colonnier, 1985), 0.8-fold greater in the monkey visual cortex compared with the mouse (Cragg, 1967), 1.4-fold greater in the human compared with the mouse (DeFelipe et al., 2002), 1.7-fold greater in the human compared with the macaque (Elston et al., 2001), 1.7-fold greater in the human compared with the rat (DeFelipe et al., 2002), and 4.6-fold greater in the monkey motor cortex compared with the mouse (Cragg, 1967). The median ratio, 1.5 ± 0.6 , is considerably smaller than the more than 1000-fold range in brain volumes represented by these species. Thus, on average, the number of synapses per neocortical neuron increases weakly or does not change as a function of brain size. (Wang et al., 2016, p. 59)

Are Wang et al. (2016) simply incorrect about the facts? If they are not, is there some way for synapse per neuron to matter greatly to intelligence within the human species while hardly varying across mammalian species?

Third, and relatedly, Herculano-Houzel (2016) stated that within primates, the scaling relationship between body size and neuron size has been abolished, such that larger primates do not have larger neurons than smaller primates. If this is true, is it still the case that humans have larger dendrites (not necessarily neurons) than macaques, say?

Reply: We would like to again apologize to the reviewer that our explanation created confusion, and we regret that this led to a misunderstanding of our findings. The role of dendrite size and synapse number in IQ scores cannot be resolved by comparing the human brain to other animal species. Within the human brain, there already is a wide variety of dendrite sizes and synapse number per neuron between brain regions. Moreover, the tests typically used to assess IQ can only be taken successfully by human participants. For these reasons, we are focusing only on human neurons in the higher-order association area MTG of which both structure and function have repeatedly been shown to be associated to human intelligence as measured with IQ tests (Deary, Penke, and Johnson 2010). Our previous work builds on that association and addressed the question whether in human MTG that has been associated with IQ scores, dendrite size and complexity and function of neurons is associated with individual variation in IQ scores between people, a logical follow up question to the MRI results published (Deary, Penke, and Johnson 2010). Moreover, there is a large body of evidence in the literature that the complexity and length of dendrites in human L3 pyramidal neurons is related to cognitive function in the human brain: in areas related to cognition, dendrites and synapse counts are increased compared to primary sensory areas (Elston, Benavides-Piccione, and DeFelipe 2001; Elston and Fujita 2014; Elston 2003; Wagstyl et al. 2015), in dementia dendritic trees are less extensive than in adult human brains (Buell and Coleman 1979) and L3 pyramidal neurons are especially vulnerable and disappear (Hof, Morrison, and Cox 1990). Finally, our own studies provide evidence that TDL in MTG correlates with IQ in humans (Goriounova et al. 2018; see reviewed in Galakhova et al. 2022).

9. Confusion between single-nucleotide polymorphisms (SNPs) and protein-coding genes. The authors repeatedly confuse SNPs (see any good textbook or online resource for a definition) and genes (“cistrons”), particularly when making some point about the small effects of common SNPs on quantitative phenotypes. For example, this sentence on p. 3 makes no sense: “GWAS typically shows extremely small effect sizes for each gene (less than 0.1% [5]), and only all associated genes together explain sufficient amount of variation in cognitive ability between individuals.” It makes no sense because it is really talking about single-nucleotide polymorphisms (SNPs), not genes. Consequently, the rest of the paragraph does not follow. Please start the paragraph with “To investigate the gene-expression profiles across areas and neurons ...”
Reply: we deleted this sentence as the reviewer suggested.

There is a similar confusion on p. 11. “Although the reported effect sizes for each gene in GWAS are extremely small [5,6], small genetic effects may have large consequences for brain function and cognitive ability.” And even besides the error in terminology, this sentence is hard to understand. What does it mean for a small effect to have a large consequence? Does this mean that the effect of a SNP on the mediator in the causal chain from genotype to years of education may not be as small? Whatever the meaning, it must be made clearer.

Reply: We removed this sentence from the text and altered the paragraph to: “Our results show that cellular phenotypes associated with intelligence - dendritic size and AP rise speed – may be influenced by genes associated with cognitive ability. Are these genes also related to human evolution? For this, we focused on the subset of IQ, EA and HAR genes that significantly correlated with either TDL (gene names listed in Fig 3E) or AP rise speed (gene names listed in Fig 4G) and were at the intersection between the IQ, EA and HAR gene sets - belonged to at least two gene sets. These genes might be especially important since their expression is not only related to cellular function (AP rise speed) and morphological structure (TDL) but they are also implicated both in human cognition and brain evolution. This selection resulted in a set of 42 TDL-correlated and 33 AP rise speed-correlated genes. Since several genes correlated both with TDL and AP rise speed, the total gene set of interest comprised 62 unique genes (Figure 5A, names of the genes are listed to the right). We investigated the function of this gene set of interest using GeneCards – the human gene database [35,36]. Importantly, almost all of these genes were protein coding and were involved in several brain related disorders (Table S1, page 14, lines 23-33).

If there any other mix-ups of this kind (there is another on p. 14), they must be fixed as well.

Reply: We have attempted to catch all other mix-ups and corrected them, but given the little detail provided, we cannot be fully certain that we caught all mix-ups the reviewer may be referring to here.

10. Errors in abbreviations for gene names. In the main text the authors refer to GRIM3 and GRIM8, but what they really mean are *GRM3* and *GRM8*. Similarly, they say KCNIMA1 when they really mean *KCNMA1*. Please double-check all gene abbreviations to be sure they are consistent with the recommendations of the HUGO Gene Nomenclature Committee. Also, it is the convention to italicize the abbreviations of gene names in *Homo sapiens*.

Reply: The names of the genes were corrected in the paper.

COMMENTS ABOUT SPECIFIC PARTS OF THE PAPER

11. Key Findings: “Genes associated with intelligence, educational attainment and HARs identified by GWAS are enriched in higher-order association cortical areas of the human brain associated with IQ scores” What does this mean? I think it has to be reworded anyway because surely HARs are not identified using GWAS.

Reply: We changed this sentence.

12. p. 2: “neurofilament marker SMI-32 marking neurons” Repetition of “mark” seems clumsy.

Reply: “marking” is changed to “staining”.

13. p. 7: “(Fig B-D)” → “(Fig 3B-D)”

Reply: this part of the text is deleted

14. p. 14: “GWAS identify thousands of single nucleotide polymorphisms only a small proportion of which is directly located in exonic, protein-coding regions (<2%) while majority are intergenic and intronic.” This sentence needs to be rewritten. It is missing a comma and an article. Also, please check the figure of 2 percent. While it is certainly true that most lead SNPs identified in GWAS are not located in exons, I think the figure may be more than 2 percent. In the absence of an authoritative source, it is better to refrain from spurious precision.

Reply: We changed this sentence.

15. p. 14: “We find that majority of GWAS-identified genes of cognition are expressed in these adult neurons and their expression is elevated in specific types. That points to a continuous functional role of these genes in human cognition rather than exclusively during critical developmental periods.” Again, there is a missing article before the word “majority.” More importantly, the entire sentence seems unjustified. Perhaps I missed something, but where in this paper is there support for the claim that a majority of the members of a GWAS gene set are expressed above some threshold in a specific type of neuron? If I am correct to think that such a quantitative claim is unsupported, then the second sentence must be stricken.

Reply: we rephrased these sentences on p 17 lines 44-47.

REFERENCES

Benjamin, D.J. et al. (2018). Redefine statistical significance. *Nature Human Behaviour*, 2, 6-10.

Finucane, H.K. et al. (2018). Heritability enrichment of specifically expressed genes identifies disease-relevant tissues and cell types. *Nature Genetics*, 50, 621-629.

Herculano-Houzel, S. (2016). *The human advantage: A new understanding of how our brain became remarkable*. MIT Press.

Lee, J.J. et al. (2018). Gene discovery and polygenic prediction from a genome-wide association study of educational attainment in 1.1 million individuals. *Nature Genetics*, 50, 1112-1121.

Okbay, A. et al. (2016). Genome-wide association study identifies 74 loci associated with educational attainment. *Nature*, 533, 539-542.

Okbay, A. et al. (2022). Polygenic prediction of educational attainment within and between families from genome-wide association analyses in 3 million individuals. *Nature Genetics*, 54, 437-449.

Reed, T.E., & Jensen, A.R. (1991). Arm nerve conduction velocity (NCV), brain NCV, reaction time, and intelligence. *Intelligence*, 15, 33-47.

Savage, J.E. et al. (2018). Genome-wide association meta-analysis in 269,867 individuals identifies new genetic and functional links to intelligence. *Nature Genetics*, 50, 912-919.

Sporns, O. (2011). *Networks of the brain*. MIT Press.

Stetson, D.S., Albers, J.W., Silverstein, B.A., & Wolfe, R.A. (1992). Effects of age, sex, and anthropometric factors on nerve conduction measures. *Muscle and Nerve*, 15, 1095-1104.

Wang, S. S.-H., Ambrosini, A.E., & Wittenberg, G.M. (2016). Evolution and scaling of dendrites. In Stuart, G., Spruston, N., & Hausser, M. (Eds.), *Dendrites* (3rd ed.) (pp. 47-75). Oxford University Press.

Reviewer #3:

It was a pleasure to read this intriguing article where the authors provide evidence that 3 different, but significantly overlapping gene sets (educational attainment, HARs, and IQ) are enriched within brain regions associated with IQ. They follow-up on this finding by investigating the correlation of gene expression patterns from single cell RNA sequencing and Patch-sequencing. Together, these data provide evidence that their overlapping gene-sets are highly correlated with dendritic length. As the authors state, such a correlation helps to understand the potentially shared mechanisms of human brain evolution and the development of human IQ. The current manuscript provides interesting findings with that likely advance the field of human brain evolution. No Major concerns.

Reply: We thank the reviewer for their support.

Minor Suggestion:

There has been a lot of interest in the types of genes studied by these authors, but typically prior studies have focused on evolutionary genetics, population genetics (GWAS), or disease associated variants. This study provides a unique assessment of the potential mechanisms by which these genes may contribute to human IQ. The one limitation to the study is that the correlation to IQ associated regions of the brain does not necessarily imply causality. The current study could benefit by adding additional evidence that the regions/genes do cause/contribute to IQ-related features in humans through additional inclusion and discussion of GWAS data and, possibly, more importantly, human disease. It is known that many of the genes they included in their analyses cause a range of neurodevelopmental disorders. It seems that they could strength the case that these genes and the associated dendritic length differences underlie intelligence, which as a term is also somewhat subjective, by including analyses of gene-disease associations. Even more, many of the associated disorders are well-studied in both humans and animal models. Therefore, it is possible that a more conclusive link to specific phenotypes like dendritic length and IQ could be ascertained from the inclusion of data from prior disease studies.

Reply: We are grateful to the reviewer for this suggestion. We have included a table with the full list of all TDL and AP correlated genes that are part of two or all three gene sets. We added the annotations of all these genes and the diseases associated with each of these genes. Most of the identified genes are also involved in a range of neurodevelopmental disorders (Table S1) marked by changes in cognitive function: including schizophrenia, autism, epilepsy, depression, mental retardation and intellectual developmental disorder. We also added text to the manuscript mentioning this finding (p 18 lines 21-24).

References

- Berg, Jim, Staci A. Sorensen, Jonathan T. Ting, Jeremy A. Miller, Thomas Chartrand, Anatoly Buchin, Trygve E. Bakken, et al. 2021. "Human Neocortical Expansion Involves Glutamatergic Neuron Diversification." *Nature* 598 (7879): 151–58. <https://doi.org/10.1038/s41586-021-03813-8>.
- Buell, S J, and P D Coleman. 1979. "Dendritic Growth in the Aged Human Brain and Failure of Growth in Senile Dementia." *Science (New York, N.Y.)* 206 (4420): 854–56. <http://eutils.ncbi.nlm.nih.gov/entrez/eutils/elink.fcgi?dbfrom=pubmed&id=493989&retmode=ref&cmd=prlinks>.
- Choi, Yu Yong, Noah A Shamosh, Sun Hee Cho, Colin G DeYoung, Min Joo Lee, Jong-Min Lee, Sun I Kim, et al. 2008. "Multiple Bases of Human Intelligence Revealed by Cortical Thickness and Neural Activation." *The Journal of Neuroscience : The Official Journal of the Society for Neuroscience* 28 (41): 10323–29. <http://www.jneurosci.org/cgi/doi/10.1523/JNEUROSCI.3259-08.2008>.
- Colom, Roberto, Richard J Haier, Kevin Head, Juan Álvarez-Linera, María Ángeles Quiroga, Pei Chun Shih, and Rex E Jung. 2009. "Gray Matter Correlates of Fluid, Crystallized, and Spatial Intelligence: Testing the P-FIT Model." *Intelligence* 37 (2): 124–35. <http://linkinghub.elsevier.com/retrieve/pii/S0160289608000925>.
- Deary, Ian J, Lars Penke, and Wendy Johnson. 2010. "The Neuroscience of Human Intelligence Differences." *Nature Reviews. Neuroscience* 11 (3): 201–11. <http://www.nature.com/doi/10.1038/nrn2793>.
- DeFelipe, Javier. 2011. "The Evolution of the Brain, the Human Nature of Cortical Circuits, and Intellectual Creativity." *Frontiers in Neuroanatomy* 5 (January): 29. <http://journal.frontiersin.org/article/10.3389/fnana.2011.00029/abstract>.
- DeFelipe, Javier, Lidia Alonso-Nanclares, and Jon I Arellano. 2002. "Microstructure of the Neocortex: Comparative Aspects." *Journal of Neurocytology* 31 (3–5): 299–316. <http://eutils.ncbi.nlm.nih.gov/entrez/eutils/elink.fcgi?dbfrom=pubmed&id=12815249&retmode=ref&cmd=prlinks>.
- Doan, Ryan N, Byoung-Il Bae, Beatriz Cubelos, Cindy Chang, Amer A Hossain, Samira Al-Saad, Nahit M Mukaddes, et al. 2016. "Mutations in Human Accelerated Regions Disrupt Cognition and Social Behavior." *Cell* 167 (2): 341-354.e12. <https://linkinghub.elsevier.com/retrieve/pii/S0092867416311692>.
- Elston, Guy N., Ruth Benavides-Piccione, and Javier DeFelipe. 2001. "The Pyramidal Cell in Cognition: A Comparative Study in Human and Monkey." *The Journal of Neuroscience* 21 (17): RC163. <https://doi.org/10.1523/JNEUROSCI.21-17-j0002.2001>.
- Elston, Guy N. 2003. "Cortex, Cognition and the Cell: New Insights into the Pyramidal Neuron and Prefrontal Function." *Cerebral Cortex (New York, N.Y. : 1991)* 13 (11): 1124–38. <https://academic.oup.com/cercor/article-lookup/doi/10.1093/cercor/bhg093>.
- Elston, Guy N, and Ichiro Fujita. 2014. "Pyramidal Cell Development: Postnatal Spinogenesis, Dendritic Growth, Axon Growth, and Electrophysiology." *Frontiers in Neuroanatomy* 8 (2): 78. <http://journal.frontiersin.org/article/10.3389/fnana.2014.00078/abstract>.
- Eyal, Guy, Huibert D Mansvelde, Christiaan P J de Kock, and Idan Segev. 2014. "Dendrites Impact the Encoding Capabilities of the Axon." *The Journal of Neuroscience : The Official Journal of the Society for Neuroscience* 34 (24): 8063–71. <http://www.jneurosci.org/cgi/doi/10.1523/JNEUROSCI.5431-13.2014>.
- Galakhova, A.A., S. Hunt, R. Wilbers, D.B. Heyer, C.P.J. de Kock, H.D. Mansvelde, and N.A. Goriounova. 2022. "Evolution of Cortical Neurons Supporting Human Cognition." *Trends in Cognitive Sciences* 26 (11): 909–22. <https://doi.org/10.1016/j.tics.2022.08.012>.
- Goriounova, Natalia A, Djai B Heyer, René Wilbers, Matthijs B Verhoog, Michele Giugliano, Christophe Verbist, Joshua Obermayer, et al. 2018. "Large and Fast Human Pyramidal Neurons Associate with Intelligence." *ELife* 7 (December): e41714. <https://elifesciences.org/articles/41714>.
- Heyer, D B, R Wilbers, A A Galakhova, E Hartsema, S Braak, S Hunt, M B Verhoog, et al. 2022. "Verbal and General IQ Associate with Supragranular Layer Thickness and Cell Properties of the Left Temporal Cortex." *Cerebral Cortex* 32 (11): 2343–57. <https://doi.org/10.1093/cercor/bhab330>.
- Hof, Patrick R., John H. Morrison, and Kevin Cox. 1990. "Quantitative Analysis of a Vulnerable Subset of Pyramidal Neurons in Alzheimer's Disease: I. Superior Frontal and Inferior Temporal Cortex." *The Journal of Comparative Neurology* 301 (1): 44–54. <https://doi.org/10.1002/cne.903010105>.
- Hutsler, Jeffrey J., Dong Geun Lee, and Kristin K. Porter. 2005. "Comparative Analysis of Cortical Layering and Supragranular Layer Enlargement in Rodent Carnivore and Primate Species." *Brain Research* 1052 (1): 71–81. <https://doi.org/10.1016/j.brainres.2005.06.015>.
- Ilin, Vladimir, Aleksey Malyshev, Fred Wolf, and Maxim Volgushev. 2013. "Fast Computations in Cortical Ensembles Require Rapid Initiation of Action Potentials." *The Journal of Neuroscience : The Official Journal of the Society for Neuroscience* 33 (6): 2281–92. <http://www.jneurosci.org/cgi/doi/10.1523/JNEUROSCI.0771-12.2013>.

- Karama, S, Y Ad-Dab'bagh, R J Haier, I J Deary, O C Lyttelton, C Lepage, and A C Evans. 2009. "Positive Association between Cognitive Ability and Cortical Thickness in a Representative US Sample of Healthy 6 to 18 Year-Olds." *Intelligence* 37 (2): 145–55. <http://linkinghub.elsevier.com/retrieve/pii/S0160289608001074>.
- Lee, James J., Robbee Wedow, Aysu Okbay, Edward Kong, Omeed Maghzian, Meghan Zacher, Tuan Anh Nguyen-Viet, et al. 2018. "Gene Discovery and Polygenic Prediction from a Genome-Wide Association Study of Educational Attainment in 1.1 Million Individuals." *Nature Genetics* 50 (8): 1112–21. <https://doi.org/10.1038/s41588-018-0147-3>.
- Savage, Jeanne E, Philip R Jansen, Sven Stringer, Kyoko Watanabe, Julien Bryois, Christiaan A de Leeuw, Mats Nagel, et al. 2018. "Genome-Wide Association Meta-Analysis in 269,867 Individuals Identifies New Genetic and Functional Links to Intelligence." *Nature Genetics* 50 (7): 912–19. <http://www.nature.com/articles/s41588-018-0152-6>.
- Testa-Silva, Guilherme, Matthijs B. Verhoog, Daniele Linaro, Christiaan P. J. de Kock, Johannes C. Baayen, Rhiannon M. Meredith, Chris I. De Zeeuw, Michele Giugliano, and Huibert D. Mansvelder. 2014. "High Bandwidth Synaptic Communication and Frequency Tracking in Human Neocortex." Edited by Idan Segev. *PLoS Biology* 12 (11): e1002007. <https://doi.org/10.1371/journal.pbio.1002007>.
- Volgushev, Maxim. 2016. "Cortical Specializations Underlying Fast Computations." *Neuroscientist* 22 (2): 145–64. <https://doi.org/10.1177/1073858415571539>.
- Wagstyl, Konrad, Stéphanie Larocque, Guillem Cucurull, Claude Lepage, Joseph Paul Cohen, Sebastian Bludau, Nicola Palomero-Gallagher, et al. 2020. "BigBrain 3D Atlas of Cortical Layers: Cortical and Lamina Thickness Gradients Diverge in Sensory and Motor Cortices." Edited by Henry Kennedy. *PLOS Biology* 18 (4): e3000678. <https://doi.org/10.1371/journal.pbio.3000678>.
- Wagstyl, Konrad, Lisa Ronan, Ian M. Goodyer, and Paul C. Fletcher. 2015. "Cortical Thickness Gradients in Structural Hierarchies." *NeuroImage* 111 (May): 241–50. <https://doi.org/10.1016/j.neuroimage.2015.02.036>.
- Wei, Yongbin, Siemon C de Lange, Lianne H Scholtens, Kyoko Watanabe, Dirk Jan Ardesch, Philip R Jansen, Jeanne E Savage, et al. 2019. "Genetic Mapping and Evolutionary Analysis of Human-Expanded Cognitive Networks." *Nature Communications* 10 (1): 4811–39. <http://www.nature.com/articles/s41467-019-12764-8>.

REVIEWERS' COMMENTS

Reviewer #1 (Remarks to the Author):

My concerns have been largely addressed. I appreciate the authors' extra work.

Christopher Patzke

Reviewer #2 (Remarks to the Author):

I admit to finding the revisions of this paper hard to understand. There seem to be numerous errors and infelicities scattered throughout the paper.

Here is my position. Since the paper does seem to make an advance, I recommend that the editor accept it after more revisions. The authors should take as much care with the writing of the paper as with the collection of the Patch-Seq data. That is, they should correct all of the hopefully minor mistakes and make the paper more comprehensible. I also ask that the paper not be sent out for another round of review. I do not have the time to serve as a copyeditor for these authors.

What follows is just a sample of issues:

Lines 63-65: I singled out the original sentence in my first review. The new version is hard to understand. It puts the verbs “implicated” and “correlated” in close succession, in a fashion that is technically grammatical, but difficult to parse. It repeats the word “intelligence” in close succession, perhaps referring to slightly different things each time, but nevertheless in a way that seems tautological. I find it hard to believe that a sentence revised specifically in response to a reviewer’s comment can be crafted so unartfully. Even a reversion to the original would be better than this.

Lines 126-137: I do not understand the verbal explanation here. The authors should consider whether the exposition can be improved. But looking at Figures 1 and S1, I can see that my original concern has been addressed. The variation between donors is indeed negligible in comparison to the variation within donors, and therefore even a naive statistical approach should not be misleading. This is my first encounter with the Friedman test; as I understand it, this is a non-parametric repeated-measures analysis of variance, which seems to fit the bill for testing differences among cell types when the same individuals contribute cells of all types.

Supplementary Figure S1: Shouldn’t the legend in the IQ panel say “Donor 1, Donor 2, Donor 3?” When such obvious errors are overlooked, confidence in the entire paper is undermined.

Supplementary Figure S2: What is meant by “as in Figure 1C?” Figure 1C is totally different. There must be some mistake; maybe the authors meant to say Figure 2C.

Figure 3: I'm a bit taken back by the analysis being done so differently. But I guess that's fine. Is there no longer any concern over the nested nature of the data, i.e., potential non-independence? It seems that for the sake of consistency it should be addressed in every analysis.

Figure 5: I wonder if this overrepresentation analysis should be banished to supplementary material. The analyses of TDL and AP rise speed aspire to provide much deeper and more specific information than the overrepresentation analysis can do. Yet the structure of the paper, which uses the TDL and AP analyses to filter the input to the overrepresentation analysis, implies that it is the overrepresentation analysis providing the grand finale. The overrepresentation analysis seems to me more like a sanity check. Many of these gene sets or very similar ones have been found to be significantly enriched in the GWAS of IQ and years of education.

Lines 427-428: Isn't it incorrect to call the HAR genes "GWAS-identified?" Didn't I point to this mistake in my previous review?

Reviewer #3 (Remarks to the Author):

Again, it was a pleasure to read this intriguing article. It appears the authors have made substantial improvements to the manuscript and have adequately addressed my minor concerns. I have no outstanding major or minor concerns.

REVIEWERS' COMMENTS

Reviewer #1 (Remarks to the Author):

My concerns have been largely addressed. I appreciate the authors' extra work.
Christopher Patzke

Reply: We thank the reviewer for their time and valuable feedback.

Reviewer #2 (Remarks to the Author):

I admit to finding the revisions of this paper hard to understand. There seem to be numerous errors and infelicities scattered throughout the paper.

Here is my position. Since the paper does seem to make an advance, I recommend that the editor accept it after more revisions. The authors should take as much care with the writing of the paper as with the collection of the Patch-Seq data. That is, they should correct all of the hopefully minor mistakes and make the paper more comprehensible. I also ask that the paper not be sent out for another round of review. I do not have the time to serve as a copyeditor for these authors.

Reply: we thank the reviewer for recommending the manuscript for acceptance after revision. We have once again carefully read the manuscript and clarified the text as the reviewer suggests below.

What follows is just a sample of issues:

Lines 63-65: I singled out the original sentence in my first review. The new version is hard to understand. It puts the verbs “implicated” and “correlated” in close succession, in a fashion that is technically grammatical, but difficult to parse. It repeats the word “intelligence” in close succession, perhaps referring to slightly different things each time, but nevertheless in a way that seems tautological. I find it hard to believe that a sentence revised specifically in response to a reviewer’s comment can be crafted so unartfully. Even a reversion to the original would be better than this.

Reply: We have changed this sentence to more clearly describe the referenced literature.

Lines 126-137: I do not understand the verbal explanation here. The authors should consider whether the exposition can be improved. But looking at Figures 1 and S1, I can see that my original concern has been addressed. The variation between donors is indeed negligible in comparison to the variation within donors, and therefore even a naive statistical approach should not be misleading. This is my first encounter with the Friedman test; as I understand it, this is a non-parametric repeated-measures analysis of variance, which seems to fit the bill for testing differences among cell types when the same individuals contribute cells of all types.

Reply: we clarified the explanation in this part of the manuscript text. We agree that Friedman test is the best fit for our analysis.

Supplementary Figure S1: Shouldn't the legend in the IQ panel say "Donor 1, Donor 2, Donor 3?" When such obvious errors are overlooked, confidence in the entire paper is undermined.

Reply: we corrected the legend in the IQ panel of Figure S1.

Supplementary Figure S2: What is meant by "as in Figure 1C?" Figure 1C is totally different. There must be some mistake; maybe the authors meant to say Figure 2C.

Reply: we corrected the legend in Figure S2

Figure 3: I'm a bit taken back by the analysis being done so differently. But I guess that's fine. Is there no longer any concern over the nested nature of the data, i.e., potential non-independence? It seems that for the sake of consistency it should be addressed in every analysis.

Reply: we have already addressed the nested nature of the data displayed in Figure 3 in the Supplementary Figure 2B, where we show the same data as in Figure 3A but averaged per donor.

Figure 5: I wonder if this overrepresentation analysis should be banished to supplementary material. The analyses of TDL and AP rise speed aspire to provide much deeper and more specific information than the overrepresentation analysis can do. Yet the structure of the paper, which uses the TDL and AP analyses to filter the input to the overrepresentation analysis, implies that it is the overrepresentation analysis providing the grand finale. The overrepresentation analysis seems to me more like a sanity check. Many of these gene sets or very similar ones have been found to be significantly enriched in the GWAS of IQ and years of education.

Reply: The strength of our overrepresentation analysis is that it is run only on the genes that emerge from the analyses in Figures 3 and 4, as significantly correlated with cellular properties in specific cell types. This analysis indeed provides a much deeper and more specific information on the possible function of these genes and is an important part of our paper.

Lines 427-428: Isn't it incorrect to call the HAR genes "GWAS-identified?" Didn't I point to this mistake in my previous review?

Reply: we disambiguated this difference between GWAS-identified and HAR genes in the text.

Reviewer #3 (Remarks to the Author):

Again, it was a pleasure to read this intriguing article. It appears the authors have made substantial improvements to the manuscript and have adequately addressed my minor concerns. I have no outstanding major or minor concerns.

Reply: We thank this reviewer for their time and valuable feedback.